


# Sentinel-1 SAR-based Globally Distributed Co-Seismic Landslide Detection by Deep Neural Networks

Lorenzo Nava [1,2,3], Alessandro Mondini [4], Kushanav Bhuyan [5,3], Chengyong Fang [5], Oriol Monserrat [6], Alessandro Novellino [7], and Filippo Catani [3]

[1]Department of Earth Sciences, University of Cambridge, Cambridge, UK
[2]Department of Geography, University of Cambridge, Cambridge, UK
[3]Machine Intelligence and Slope Stability Laboratory, Department of Geosciences, University of Padova, Padova, Italy
[4]National Research Council, Research Institute for Applied Mathematics and Information Technologies, Genova, Italy
[5]State Key Laboratory of Geohazard Prevention and Geoenvironment Protection, University of Technology, Chengdu, China
[6]Department of Remote Sensing, Geomatics Research Unit, Centre Tecnologic de Telecomunicacions de Catalunya (CTTC), Barcelona, Spain
[7]British Geological Survey, Keyworth, Nottingham, UK

**Correspondence:** Lorenzo Nava (ln413@cam.ac.uk)

**Abstract.** Rapid response to multiple landslide events (MLEs) demands accurate, all-weather, day-and-night detection capabilities. Optical remote sensing has advanced landslide detection but remains limited under adverse weather and lighting conditions. Synthetic Aperture Radar (SAR), resilient to these constraints, remains underexplored for automated landslide detection due to challenges such as complex pre-processing and geometric distortions. This study integrates Deep Neural Networks (DNNs) with SAR backscatter data for co-seismic landslide detection, utilizing a data-centric approach. We inform the

models using 11 earthquake-induced MLEs, covering ≈ 73,000 landslides across diverse geologic and climatic settings. Inference on unseen MLEs in Haiti (2021) and Sumatra (2022) demonstrates robust transferability, achieving F1-scores up to 82%. Using explainable artificial intelligence, we highlight the discriminative capability of change detection bands over backscatter alone. Our findings emphasize the potential of SAR-based DNN models for worldwide, generalized, and rapid landslide

detection, addressing critical gaps in current methods that solely use optical data. This research lays a foundation for broader applications in automated SAR-based earth surface change detection, particularly in complex, hilly and mountainous terrains.

## 1 Introduction

Slope instabilities, commonly referred to as landslides, represent a widespread natural occurrence in mountainous and hilly regions, presenting substantial threats to both human lives and infrastructure (Froude and Petley, 2018). Earthquakes, heavy

rainfall, and human activities serve as the primary triggers for landslides (Ferrario, 2019; Serey et al., 2019; Song et al., 2019; Wang et al., 2019). Notably, a single event can involve either one or multiple landslide failures (Guzzetti et al., 2012), commonly referred to as multiple landslide events (MLEs). Over the last 15 years, several co-seismic MLEs have occurred, impacting wide regions within remarkably brief timeframes (Tanyaş et al., 2017). In the aftermath of such occurrences, it is important to investigate locations of landslides to assess damages to natural and anthropogenic landscapes. Williams (Williams





et al., 2018) and Amatya (Amatya et al., 2023) highlighted the importance of rapid mapping in disaster responce using as disasters the 2015 Gorkha and 2021 Haiti MLE co-seismic landslides. Moreover, a comprehensive understanding of these slope instability processes begins with a spatial assessment of the slope failures both for rainfall (Nocentini et al., 2023; Segoni et al., 2014) and earthquake-induced landslides (Meena and Piralilou, 2019). Information concerning the location and timing of failed slopes is usually recorded in landslide inventories (Eeckhaut et al., 2013). Traditionally, the accurate

positioning for landslides involved mainly the Global Satellite Navigation System (GNSS), topographic total station, and aerial photogrammetry surveys, known for their accuracy. However, these surveys are expensive, pose risks in challenging terrains (Manconi et al., 2014), and are exceptionally time-consuming. However, time is key during damage assessments of MLE affecting large areas (Williams et al., 2018). As a result, satellite products have emerged as a cost-effective alternative for generating landslide inventories (Ghorbanzadeh et al., 2019). The field of remote sensing, particularly Earth observation

(EO) imagery analysis, has been extensively investigated for automatically extracting landslide location. However, a significant portion of landslides lacks comprehensive and timely information (Guzzetti et al., 2012). This deficiency stems from the lack of systematic coverage in existing EO data (Williams et al., 2018). The challenge is exacerbated by the reliance on optical imagery. The methods that use this data type are generally precise, however, limited when solar reflection is absent, or in presence of cloud obscuration. Numerous research has recently combined optical data with deep learning (DL) models for landslide

detection (Novellino et al., 2024). These approaches range from utilizing crowdsourced data (Catani, 2021) and Unmanned Aerial Vehicles (UAVs) (Beni et al., 2023; Dai et al., 2023b) to analyzing LIDAR (Fang et al., 2022), and optical satellite imagery (Bhuyan et al., 2023; Prakash et al., 2021). Some studies have also investigated the incorporation of morphological factors alongside satellite data for DL-based landslide detection (Meena et al., 2021). Lastly, there is a growing trend towards training DL models capable of delivering reliable predictions in unseen areas for swift assessment of new MLEs. Some of

these studies concentrate on individual data sources, such as Copernicus Sentinel-2 (Prakash et al., 2021; Ghorbanzadeh et al., 2022) and PlanetScope (Meena et al., 2023), while others examine the integration of multi-source data (Xu et al., 2024; Fang et al., 2024) to enhance accuracy and improve transferability. However, persistent clouds hinder the timely acquisition of data, impeding effective disaster management operations (Mondini et al., 2021a). This issue is prevalent in numerous tropical countries and is universally present in the context of landslide activations induced by storms (Wilson and Jetz, 2016) or where,

following earthquakes that triggered MLEs, the first cloud-free optical image is not rapidly available (Williams et al., 2018). SAR sensors represent a valuable alternative due to their capacity to acquire information on the ground in all-weather and illumination conditions (Hertel et al., 2023). Recently, there has been a growing interest in using the amplitude information to identify and rapid landslide failures (Burrows et al., 2020, 2019; Catani et al., 2005; Chorowicz et al., 1998; Mondini et al., 2019; Santangelo et al., 2022; Singhroy, 1995). Landslides are identified as anomalies in SAR products based on tone, texture,

pattern, mottling, or their changes (Santangelo et al., 2022; Singhroy, 1995; Lindsay et al., 2023). Konishi and Suga (2018); Suga and Konishi (2012); Uemoto et al. (2019) utilized X-band COSMO-SkyMed and airborne Pi-SAR2 imagery, respectively, for landslide mapping in Japan.

Handwerger et al. (2022) designed a GEE-based approach to produce density heatmaps with Sentinel-1 C-band data over several MLEs. However, while most of these methods have demonstrated the ability to successfully detect landslides in in-





dividual study areas, the issue of transferability in different settings and with different satellite data persists (Mondini et al., 2021b). While artificial intelligence shows promise in training generalized models for automatically assessing target locations using satellite imagery (Guan et al., 2023), the focus on optical-based automated landslide detection outweighs studies utilizing SAR data (Mondini et al., 2021b). Challenges like data pre-processing (Plank et al., 2016) and geometry distortions, particularly in high-slope landslide-prone regions, contribute to this disparity. Instances, where SAR and DL are combined

for landslide detection, remain rare. Among these, Nava et al. (2022a, b) demonstrated promising results by employing SAR Sentinel-1 amplitude data and convolutional neural networks (CNNs) in Hokkaido, Japan. Additionally, Shi et al. (2023) evaluated automated approaches in Papua New Guinea and Milin using maximum likelihood estimation. However, these studies have focused on single-event evaluations rather than investigating scalable, globally applicable solutions. Hence, a comprehensive approach, specifically focusing on a globally distributed Sentinel-1 SAR-based landslide detection approach using deep learn-

ing, is currently missing. Our research introduces a generalized approach for rapid landslide detection, integrating Sentinel-1 SAR backscatter data and CNNs. Unlike existing methods, which often lack reproducibility across diverse geo-settings, our approach demonstrates full transferability to unseen events, informed on 11 earthquake-induced landslide events (~73,000 landslides). The method incorporates a SAR-specific data preparation approach to ensure consistent performance across varied morphologies. We enhance transparency and reliability through eXplainable AI (XAI), identifying pixel-level contributions

to understand predictions. Additionally, we open-source the SAR-based Landslide Rapid Assessment (SAR-LRA) tool that enables immediate detection of landslides, leveraging Google Earth Engine (GEE) and Google Colaboratory for cloud-based processing. To the best of our knowledge, this research is the first to achieve fully transferable SAR-based landslide detection via DNNs, addressing the critical gap in reproducibility and laying a foundation for automated, generalized SAR-based co-seismic landslide response.

## 2 Study Areas and Materials

### 2.1 Study Areas

This study investigates 11 earthquake-triggered multiple landslide events (MLEs) across diverse geographic and geologic settings. These events provide a representative dataset for assessing landslide detection performance across different environmental and tectonic contexts.

**Gorkha, Nepal, 2015 (Mw 7.8)**: The Gorkha earthquake struck Nepal on April 25, 2015, with a magnitude of Mw 7.8, caused by the convergence of the Indian and Eurasian tectonic plates. This devastating event resulted in nearly 9,000 fatalities and significant economic damage. For this study, we use the inventory provided by Roback et al. (2017), which documented 24,915 landslides over an area of 87 km$^2$. These were mapped using high-resolution satellite imagery, including WorldView-2 and -3 and Pleiades, with spatial resolutions ranging from 20–50 cm.

**Kaikōura, New Zealand, 2016 (Mw 7.8)**: The Kaikōura earthquake occurred on November 14, 2016, on the South Island of New Zealand, with an Mw of 7.8. The earthquake caused over 10,000 landslides, primarily in sparsely populated mountainous areas. The inventory by Tanyaş et al. (2022a) was used in this study, which includes 14,233 mapped landslides covering



14,000 km$^2$. The landslides were identified using Sentinel-2 satellite images at a 10 m resolution for pre- and post-earthquake conditions.

**Capellades, Costa Rica, 2016 (Mw 5.3)**: The Capellades earthquake struck Costa Rica in 2016 with a magnitude of Mw 5.3, triggering numerous landslides. The inventory used in this study was developed by Ruiz et al. (2020), utilizing field surveys, LIDAR data, drone photography, and high-resolution satellite imagery. Additionally, a detailed Digital Elevation Model (DEM) with 20 m contours was employed to map the landslides.

    **Milin, China, 2017 (Mw 6.9)**: The Milin earthquake occurred on November 18, 2017, in the Bomi-Medog structural belt of 95  southeastern Tibet, with a magnitude of Mw 6.9. We use the inventory by Hu Hu et al. (2019), which identified 939 landslides covering an area of 37.65 km$^2$. Landslides were mapped using post-event imagery from Spot 7 (1.3 m resolution) and Sentinel-2 (9.4 m resolution).

    **Papua New Guinea, 2018 (Mw 7.5)**: On February 25, 2018, a powerful Mw 7.5 earthquake struck the central highlands of Papua New Guinea, an area within the tectonically active "Ring of Fire." The event triggered 11,607 landslides, predominantly 100  in regions of high relief, steep slopes, and weak lithology. The inventory by Tanyaş et al. (2022b) was used, covering 185 km$^2$. Landslides were mapped using high-resolution satellite imagery.

    **Lombok, Indonesia, 2018 (Mw 6.3 and Mw 6.9)**: Between July and August 2018, a series of earthquakes with magnitudes exceeding Mw 6.0 occurred on Lombok Island, Indonesia. The landslides were mapped by Ferrario (Ferrario, 2019) and cover 1798 km$^2$. The inventory merges data from different earthquakes to ensure consistency.

**Hokkaido, Japan, 2018 (Mw 6.6)**: The Iburi earthquake struck Hokkaido, Japan, on September 5, 2018, with an Mw of 6.6. It triggered 7837 landslides, primarily in mountainous regions of the eastern and central Iburi areas. The landslides were mapped by Wang et al. (2019) using high-resolution PlanetScope imagery.

    **Mesetas, Colombia, 2019 (Mw 6.0)**: The Mesetas earthquake occurred on December 24, 2019, with an Mw of 6.0. It affected the eastern foothills of the Colombian Eastern Cordillera, triggering widespread landslides. The inventory by García-110  Delgado et al. (2021) was used, which was created as part of a rapid-response effort using satellite imagery.

    **Haiti, 2021 (Mw 7.2)**: On August 14, 2021, the Mw 7.2 Nippes earthquake struck Haiti, causing extensive landslides in the Tiburon Peninsula. The inventory by Martinez et al. (2021), developed as part of a rapid-response effort, was used. It includes landslides mapped using high-resolution post-event imagery.

    **Luding, China, 2022 (Mw 6.6)**: The Luding earthquake struck on September 5, 2022, with an Mw of 6.6. Located in 115  the Hengduan Mountains, the earthquake triggered 5336 landslides. The inventory by Dai et al. (2023a) was used, based on high-resolution imagery from PlanetScope, Gaofen-2, Gaofen-6, and UAV surveys.

    **Sumatra, Indonesia, 2022 (Mw 6.1)**: The Sumatra earthquake occurred on February 25, 2022, with an Mw of 6.1. It caused significant landslide activity, which was mapped using PlanetScope imagery in the methodology outlined by Meena et al. (2023). This inventory captures landslides triggered across the affected region.





**Table 1.** Sentinel-1 Polarizations and Orbits Accessible for Each Study Case

| Study Case | Polarizations | Orbits | Date of Event | Number of Landslides |
|---|---|---|---|---|
| Gorkha | VV | Ascending; Descending | 25 April 2015 | 24,903 |
| Kaikōura | VV | Ascending; Descending | 14 November 2016 | 14,168 |
| Capellades | VV | Ascending; Descending | 1 December 2016 | 51 |
| Milin | VV, VH | Ascending; Descending | 18 November 2017 | 766 |
| Papua New Guinea | VV, VH | Ascending; Descending | 26 February 2018 | 4,584 |
| Lombok | VV, VH | Ascending; Descending | 5 August 2018; 19 August 2018 | 12,688 (total) |
| Hokkaido | VV, VH | Ascending; Descending | 6 September 2018 | 5,625 |
| Mesetas | VV, VH | Ascending; Descending | 5 August 2019 | 837 |
| Haiti | VV, VH | Ascending; Descending | 14 August 2021 | 4,887 |
| Luding | VV, VH | Ascending; Descending | 5 September 2022 | 5,006 |
| Sumatra | VV, VH | Ascending; Descending | 25 February 2022 | 171 |

## 2.2 Sentinel-1 Data and Pre-Processing

The Sentinel-1 constellation offers improved revisit capabilities compared to earlier SAR missions such as ERS-1/2 and Envisat ASAR. While maintaining broad area coverage, it surpasses its predecessors by providing higher resolution and the potential for global dual-polarization coverage over landmasses. Each Sentinel-1 satellite follows a near-polar, sun-synchronous orbit with a 12-day repeat cycle, completing 175 orbits per cycle. Sentinel-1A and Sentinel-1B are positioned in the same orbital plane but with a 180° orbital phase difference. With a single satellite, global landmass mapping in Interferometric Wide swath mode occurs approximately every 12 days, while the two-satellite constellation enables a more frequent 6-day exact repeat cycle at the equator. Notably, Sentinel-1B has been inactive since 2022, and it is in the process of being substituted by an equivalent platform. Revisit rates vary with latitude, with higher revisit frequencies observed at higher latitudes compared to the equator (Sentinel Online). We used GRD scenes with 10 meters spatial resolution and Interferometric Wide (IW) acquisition mode. Depending on availability, each scene consists of either one or two out of four possible polarization bands, determined by the instrument's polarization settings. The potential combinations include single-band Vertical Vertical (VV) polarization or Vertical Horizontal (VH) polarization, and dual-band VV+VH, each representing different co-polarization or cross-polarization scenarios. Additionally, each scene incorporates an 'angle' band indicating the approximate incidence angle from the ellipsoid in degrees at each point. Pre-processing for each scene includes thermal noise removal, radiometric calibration, and terrain correction using SRTM 30 or ASTER DEM (for latitudes above 60 degrees where SRTM data is unavailable), performed via the Sentinel-1 Toolbox. The final terrain-corrected values are logarithmically scaled ($10\log_{10}(x)$ dB). For a comprehensive understanding of the pre-processing steps, refer to https://developers.google.com/earth-engine/guides/sentinel1. Our research





focuses on scenarios where either only VV polarization or both VV and VH polarizations are accessible for both ascending and descending orbits (see Table 1).

## 3  Methods

Pre- and post-event SAR imagery undergo several pre-processing steps on the GEE cloud, including acquisition, stack median calculation, differencing, and stacking. Specifically, for each study area, satellite images are acquired and stacked in space and time, median values of the stacks are extracted, and shadowing and layover masks are generated. Subsequently, change detection bands are calculated, and the final images are composited. The subsequent data processing involves image sampling to create datasets, as outlined in Section 3.1, to prepare the data for model training (see Fig. 1). This involves organizing the imagery and associated metadata into structured and reliable datasets suitable for input into the model training pipeline. The data is then fed into the CNN for calibration, as described in Section 3.3. During the landslide detection phase, the downloaded images undergo the same processing steps, ensuring uniformity in the data preprocessing procedure. Once the images are prepared, the object detection procedure outlined in Section 3.4 is applied.

### 3.1  Dataset Design

Four distinct combinations of Sentinel-1 SAR polarizations and orbits have been devised to evaluate neural network classification performance. These combinations are described in Table 2. We also calculate the differences for both pre- and post-VV and VH amplitude imagery, represented as *diffVV* and *diffVH*, respectively. These values are derived by subtracting the median amplitude of the pre-event stack from the median amplitude of the post-event stack. Each polarization combination is separately employed with different orbits, enabling the independent assessment of the neural network's classification performance.

**Table 2.** Main Dataset Combinations. *diffVV* and *diffVH* are obtained by subtracting the median of the pre-event stack from the median of the post-event stack.

| Name | Orbit | Var 1 | Var 2 | Var 3 | Var 4 |
|---|---|---|---|---|---|
| VV | Ascending | VV post-event | diffVV | / | / |
|  | Descending | VV post-event | diffVV | / | / |
| VV_VH | Ascending | VV post-event | VH post-event | diffVV | diffVH |
|  | Descending | VV post-event | VH post-event | diffVV | diffVH |

Furthermore, these four major combinations are implemented across various pre- and post-event temporal buffers. The selection of these buffers is based on the current average revisiting time of Sentinel-1. The 12-day buffer represents the most rapid assessment time, encompassing one acquisition per acquisition geometry, which will fall within 12 days of the event (Sentinel Online). We vary the selection of such event buffers to assess their impact on model performance (see Table 3).



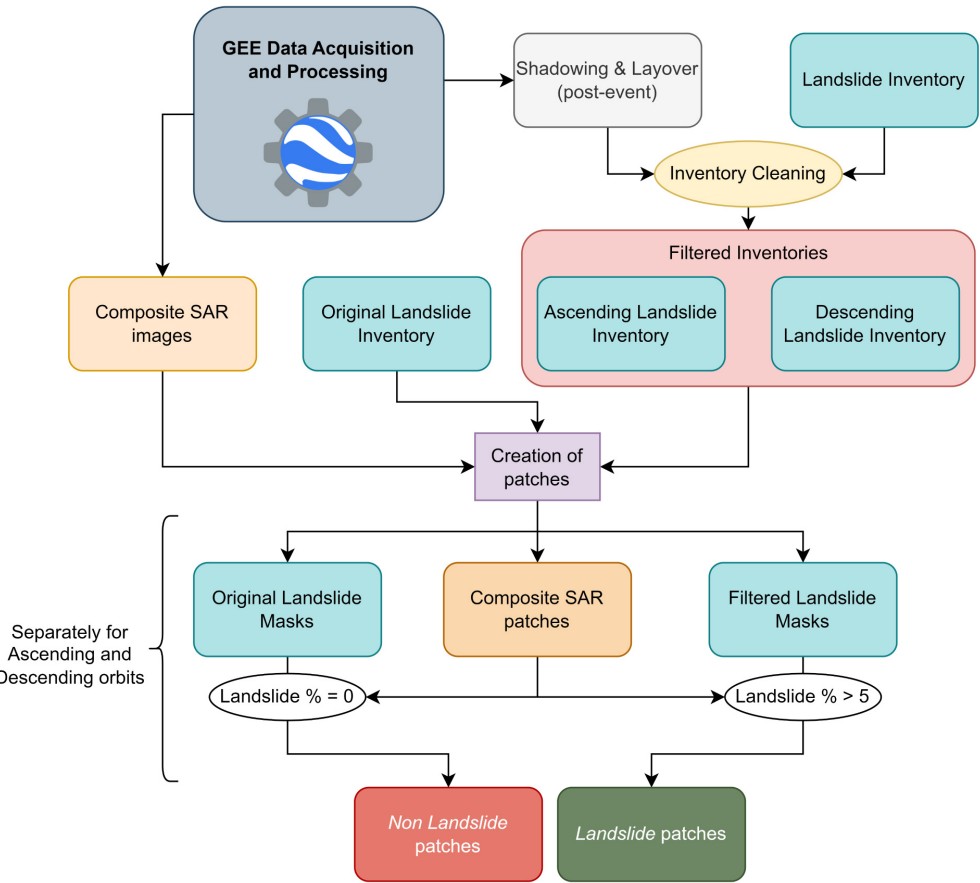

**Figure 1.** Overall workflow of the sampling strategy, model calibration, and deployment. Panel (a) illustrates the detailed sampling process of the SAR dataset, which filters out areas affected by geometric distortions based on the satellite's orbit. After filtering, patches are created, and a landslide pixel threshold of 5% is applied to ensure the quality of the dataset. This curated dataset is then used to train and test the CNN classifier. This approach is iteratively applied for each dataset combination and temporal buffer assessed in our study (see Tables 2 and 3). Once calibrated, the CNN is deployed (panel (b)) over new, unseen regions to detect co-seismic landslides.

**Table 3.** Temporal Buffers Applied to the Dataset Combinations in Table 2

| Pre-event (days) | 365 | 240 | 120 | 60 | 60 | 60 | 12 | 24 | 48 | 60 |
|---|---|---|---|---|---|---|---|---|---|---|
| Post-event (days) | 12 | 12 | 12 | 12 | 24 | 48 | 12 | 24 | 48 | 60 |

It is crucial to acknowledge that when working with two multitemporal stacks located in different geographic areas, even if they are generated using the same temporal buffers, they might not have the same number of images. This discrepancy can be attributed to differences in the orbit plans of Sentinel-1. A total of 28 datasets have been generated, encompassing all the above-mentioned combinations.





## 3.2 Data Sampling

Landslide polygons from the available inventories are de using optical data. Because of this, intrinsic differences between SAR and optical data must be considered during the sampling process to ensure accurate analysis. The primary distinctions arise from the geometric distortions inherent in SAR (Soldato et al., 2021), particularly in hilly and mountainous terrains (Burrows et al., 2020). Shadowing and layover are particularly troublesome distortions as the affected pixels in these regions cannot convey useful information. Layover represents an extreme form of foreshortening, where the upper portions of a backscattering

object, such as a mountain top, are recorded closer to the radar than the lower parts, like the base of the slope. Shadows result from areas lacking radar illumination (Meyer, 2019). The extent of geometric distortions is influenced by satellite orbit parameters (e.g., ascending or descending), satellite configuration (e.g., side of look), and acquisition parameters (e.g., look angle, Θ) (Mondini et al., 2021b). To address the potential bias introduced by including images labeled as landslides that lack relevant landslide-related information, we design an original sampling strategy reported as the "*Inventory Cleaning*" step in

Figure 1. This involves calculating shadow and layover masks specific to the acquisition geometry and study case. We then subtract the masks from the landslide inventories. The methodology for extracting shadow and layover masks aligns with the approach proposed by (Vollrath et al., 2020), as utilized by (Lindsay et al., 2022). For each study case, three landslide inventories are used: (i) original inventory, (ii) inventory filtered with ascending distortion masks, and (iii) inventory filtered with descending distortion masks. The original inventory is used for sampling the background class. Subsequently, the filtered

inventories are utilized to sample the landslide class, with the additional condition that the landslide images must comprise more than 5% of pixels classified as landslides. The threshold corresponds to an area of approximately 2000 square meters. Higher thresholds enhance model performance by enriching training and testing with more pronounced landslide signatures, yielding higher scores. However, the model may primarily detect cases with high landslide density due to the emphasis on stronger signatures. The sampling approach incorporates a grid-based patch sampling methodology, designed to avoid any

overlap between patches. The variability in landslide dimensions across different cases precludes us from selecting a patch size solely based on the landslide sizes. Generally, larger patches provide richer contextual information. However, excessively large patches yield final bounding boxes that are undesirably large. In our study, we settled on a patch size of $64 \times 64$ pixels, aiming for a final bounding box of approximately 0.4 square kilometers. Lastly, as landslide detection is a highly imbalanced binary classification problem the ratio of background/landslide in terms of the number of patches can range from approximately 8

(Hokkaido) to 120 (Gorka). Therefore, we opted to adjust the ratio of the test set to 10 in cases where the natural ratio was initially higher while maintaining it unchanged in instances where it was lower.

## 3.3 Deep Learning-Based Landslide Classification

CNNs have demonstrated remarkable effectiveness in image classification tasks (Nava et al., 2022b; Tang et al., 2021; Zhang et al., 2017; Zhou et al., 2022). For this study, we use a shallow, yet efficient CNN model for landslide detection. The model

comprises an encoder and a classifier. During the encoding phase, the input data shape is $H \times W \times C$ (Height $\times$ Width $\times$ Channels). Through three encoding modules, the data dimensions are transformed to $(H/4 \times W/4 \times 32)$. Each encoding

module consists of a convolutional layer (Conv $3 \times 3$), normalization layer, and max-pooling layer. Subsequently, to fully consider global features and mitigate the impact of feature collapse resulting from dimensionality increase, we concatenate the multi-level features from the three encoding modules. After pooling and dropout operations, the concatenated features are fed into a Dense layer for classification using the sigmoid activation function: $\sigma(x) = \frac{1}{1+e^{-x}}$. The architecture described is a modified version of the one utilized in (Nava et al., 2022b). This architecture has demonstrated high performance in image classification, including SAR-based tasks where precision is crucial (Nava et al., 2022b). We trained independent CNN models for each dataset combination, comprising 2 polarization combinations, 2 orbits (Table 2), and 10 temporal window variations (Table 3), resulting in a total of 40 trained models. The performance of the CNN, CBAM, and ResNet architectures was evaluated on the VV_VH dataset, with ResNet achieving the highest accuracy (96.43%) and F1-score (82.53%). While CBAM demonstrated competitive results, the CNN was chosen for extensive experiments due to its computational efficiency and strong baseline performance (accuracy: 96.04%, F1-score: 80.94%), making it a practical choice for large-scale testing (see Appendix and Supplementary Materials for more details). We divided the dataset into the conventional training, validation, and testing partitions (Haykin, 2009). Specifically, 67% of the data from each study area is used for training and validation, and the remaining 33% is used for testing. During model training, 40% of the training set is allocated for the validation set. The VV_VH dataset includes data from Papua New Guinea, Lombok, Hokkaido, Mesetas, Milin, and Luding. The VV dataset also incorporates data from Capellades, Kaikoura, and Gorka, as VH data is not available for these three locations (see Table 1). To evaluate model performance in new, independent scenarios, data from Sumatra and Haiti are kept completely unseen. This data is used to simulate model deployment and assess the model's generalizability. We use TensorFlow 2.8 (Abadi et al., 2016) for training the model, utilizing the Adam optimizer (Kingma and Ba, 2015) with a variable batch size and a focal loss function. The model underwent training for a maximum of 500 epochs, during which hyperparameter tuning was employed to optimize its performance. Additionally, early stopping was implemented to halt training when the validation loss ceased to decrease for 30 consecutive epochs. We use the focal loss (Lin et al., 2017), designed address the challenges of class imbalance and the effective handling of challenging examples within the context of binary classification tasks. Finally, iterative training sessions were conducted employing various hyperparameter combinations, specifically focusing on the number of filters (32, 64), class imbalance rate of the training set (ranging from 4 to 6), dropout rates (0.5, 0.7), and learning rates ($10^{-4}$, $10^{-5}$). Model performance evaluation involved standard metrics, including Precision, Recall, and F1-score, derived from true positives (TP), true negatives (TN), false positives (FP), and false negatives (FN).

## 3.4 Landslide Detection

The calibrated models are coupled with a sliding window algorithm (Lee et al., 2001) and non-maximum suppression (Neubeck and Gool, 2006) to identify landslide locations within a given area. The sliding window algorithm systematically extracts regions of interest (ROIs) across the study area, using defined vertical and horizontal strides and dimensions. In this study, the vertical and horizontal strides are set to 32 pixels, ensuring a 50% overlap between adjacent ROIs. This overlap enhances robustness by capturing diverse spatial features while maintaining computational efficiency. Furthermore, the dimensions of the training patches are fixed at $64 \times 64$ pixels to align with the characteristics of the training data. These ROIs, representing



sub-images of the satellite imagery, are classified as either landslide or non-landslide using models trained on labeled image data. To refine the predictions, non-maximum suppression is applied, eliminating redundant or overlapping detections to ensure that each classified landslide patch corresponds to a unique and significant area on the map. Although this approach is computationally more intensive compared to classic YOLO (You Only Look Once) architectures (Han et al., 2023; Liu et al., 2023),

it is preferred due to the inherent characteristics of SAR data. Unlike optical imagery, SAR data often exhibits discrepancies with landslide inventories because of factors such as shadowing, layover, and foreshortening, which can distort the appearance of landslides. These effects result in misalignments between detected landslides and inventories created using optical imagery or field surveys (Nava et al., 2022b). Additionally, landslide polygons in inventories often represent amalgamated features or multiple landslides, making it impractical to define bounding boxes based on exact polygon boundaries.

## 240 4 Results

We conduct a comparison of model predictions using VV and VV_VH data across six study areas: Papua New Guinea, Lombok, Hokkaido, Mesetas, Milin, and Luding, to ensure consistency when evaluating model performance. Additionally, the model trained on the VV datasets uses the nine cases outlined in Section 3 to improve the model's generalizability, and the corresponding performance scores are reported. Finally, we assess the models' detection capabilities in completely new and

unseen regions, simulating the detection of landslides for a recently occurred, unseen major landslide event (MLE).

### 4.1 Landslide Classification Results

As established in Section 3.1, the test sets are prepared to include a representative subset comprising 33% of patches from each distinct study case. To ensure an unbiased comparison, identical patches are employed for model training and testing phases in all the combinations. The reported scores in Tables A2 and A3 represent the median values derived from an exploration

across 24 distinct combinations of hyperparameters and training set class imbalances, as outlined in Section 3. An overview of the most meaningful metric, F1-score, across the various data configurations is given in Fig. 2. The precision trend remains relatively consistent across all combinations, while the recall varies significantly, as does the F1 score. In numerous cases, both metrics experience a significant increase when incorporating additional post-event acquisitions.

### 4.2 Landslide Rapid Assessment on Unseen MLEs

In this section, we present the detection results of the models that showcased the best rapid assessment performance for both descending and ascending orbits separately. The characteristics of the hyperparameters and performance of the models used in the rapid assessment are provided in Table 3.1. For the calibration of the VV-based models, samples from Gorka, Kaikoura, and Capellades were included in addition to those used for the VV_VH-based models. This inclusion aimed to improve the models' generalizability, albeit at the cost of slightly lower performance scores. To further assess the generalizability and reliability of

these models, we deployed them for the 2022 Sumatra event, which was not part of the initial training and test datasets. The results, illustrated in Figure 3, demonstrate the effectiveness of both the VV_VH combination models. Figure 3 compares



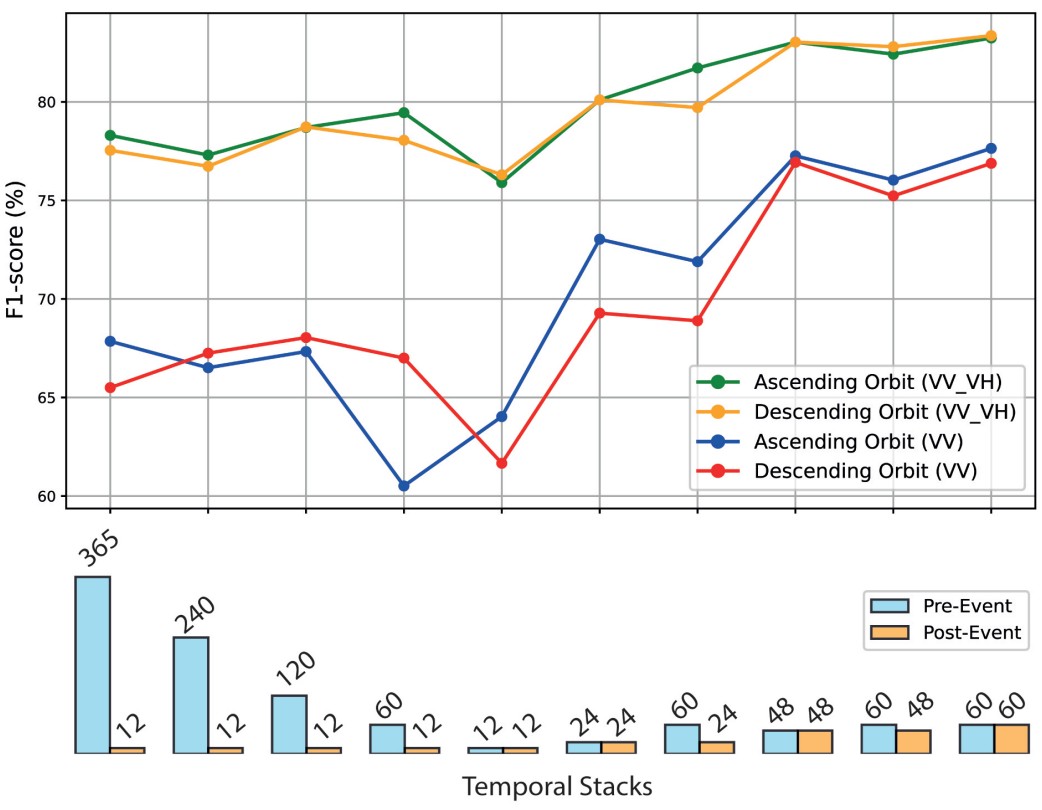

**Figure 2.** Performance of models trained on different pre- and post-event temporal stacks, considering both ascending and descending orbits and utilizing VV and VV_VH data combinations.

predictions from the VV_VH combination model across descending and ascending orbits during rapid assessment. The model maintains robust performance even when applied to novel, unseen areas and events.

**Table 4.** Hyperparameters and performance of models for rapid assessment using VV and VV_VH combinations for ascending and descending orbits. All models use a 60-day pre-event and 12-day post-event stack with a learning rate of 0.001. Lower VV scores reflect the inclusion of additional study cases (Gorka, Kaikoura, Capellades).

| Name | Orbit | Train Imbalance | Filters | Dropout Rate | Accuracy (%) | Precision (%) | Recall (%) | F1-score (%) |
|---|---|---|---|---|---|---|---|---|
| VV_VH | Ascending | 6 | 32 | 0.7 | 96.49 | 85.73 | 79.75 | 82.63 |
| | Descending | 5 | 64 | 0.5 | 96.12 | 86.27 | 74.91 | 80.19 |
| VV | Ascending | 5 | 64 | 0.5 | 90.88 | 52.55 | 63.02 | 57.31 |
| | Descending | 5 | 32 | 0.7 | 93.18 | 69.12 | 57.61 | 62.84 |





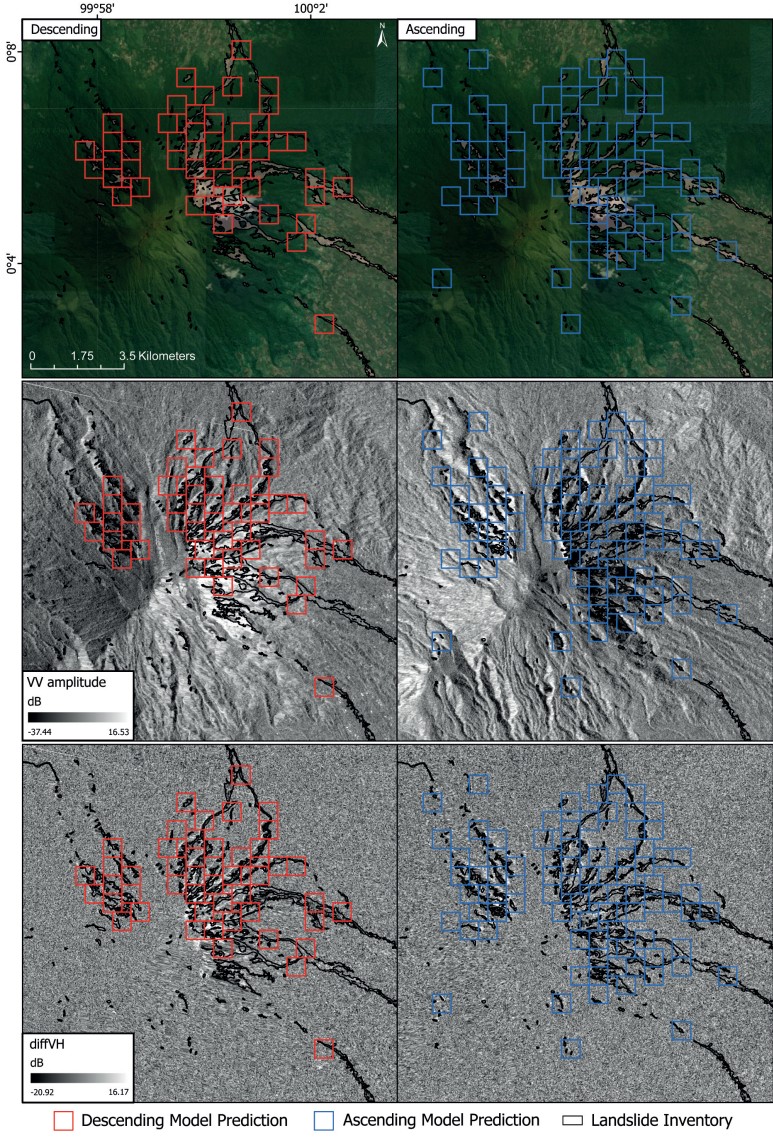

**Figure 3.** Rapid assessment for the 2022 Sumatra event utilizing VV_VH combination and 60-12 stack pre- and post-event. This unseen event validates the comprehensive generalizability in the rapid assessment of the VV_VH-based detection models. The left column shows the predictions obtained using the descending orbit, while the right shows the predictions based on the ascending orbit. The first row shows the predictions over an RGB background, the second row uses a post-event VV background, and the third row displays the diffVH background.

We further investigate the models' performance through a second unseen case study of the 2021 Haiti event, with results
presented in Figure 4. This figure highlights the strengths and limitations of the VV_VH combination model in the context of a complex event. The Haiti case is particularly challenging due to its topographical and environmental variability, which impacts model predictions. The figure reveals areas where the ascending orbit model tends to overpredict landslide occurrences.





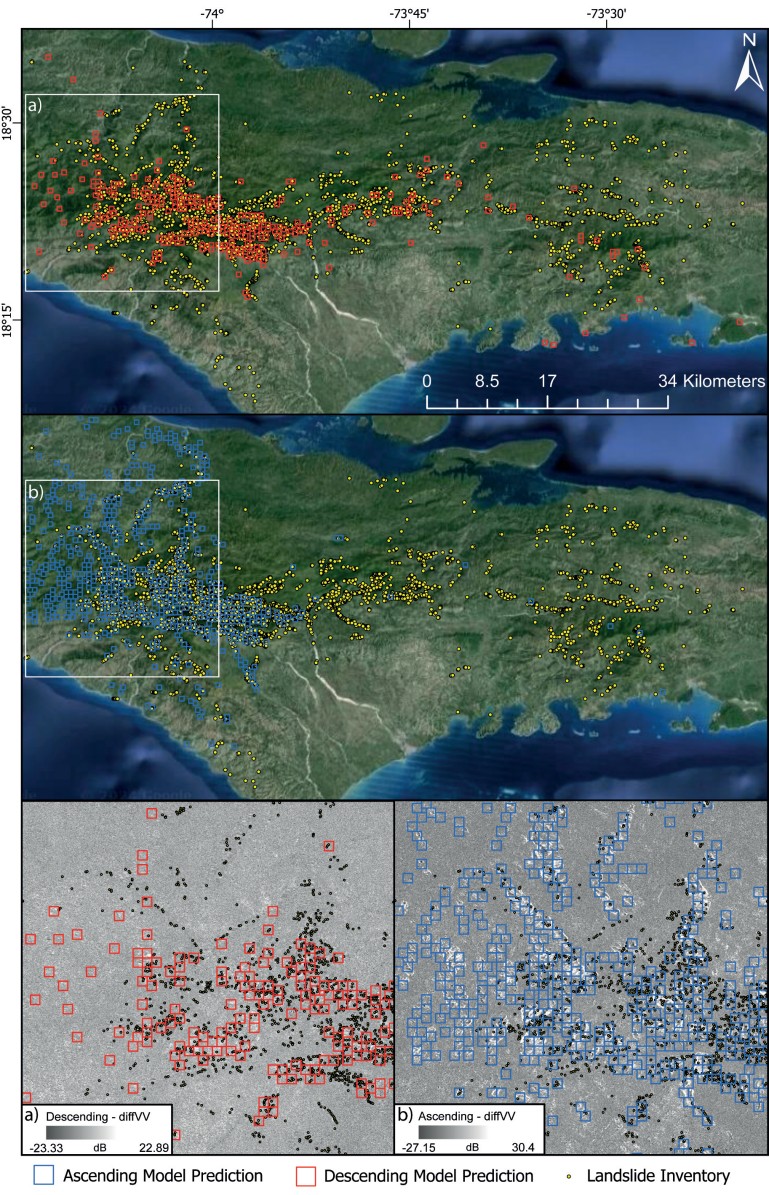

**Figure 4.** This figure presents the rapid assessment conducted for the 2021 Haiti event, utilizing the VV_VH combination and the 60-12 stack of pre- and post-event imagery, for both ascending and descending orbits. This unseen event serves as a validation of the comprehensive generalizability of the proposed models. Figures (a) and (b) depict the areas where the ascending model tends to overpredict the landslide class (FP) compared to descending-based predictions. Here the background is represented as diffVV, highlighting significant differences between pre- and post-event imagery, particularly pronounced in areas susceptible to foreshortening effects in the ascending imagery. (Source: Image Landsat / Copernicus)



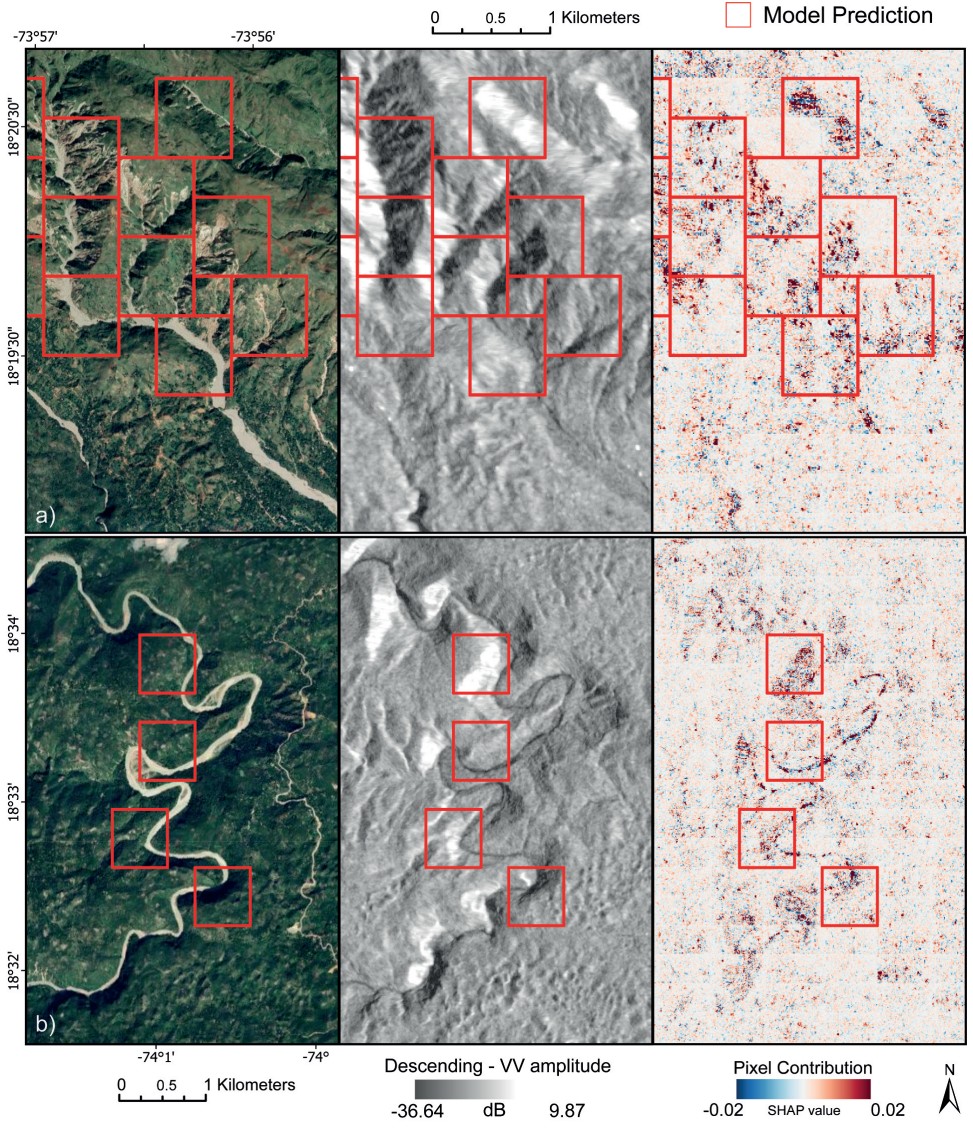

**Figure 5.** Prediction results for the 2021 Haiti event were generated using the VV_VH combination and a 60-60 stack pre- and post-event, descending orbit. a) True Positives; b) False Positives. The figure shows orthophoto, post-event VV amplitude, and Pixel Contribution (SHAP value) to derive the final detection. Notably, SHAP values represent the mean of the pixel contribution across all four bands in the image. It's important to highlight that the pixel contribution is calculated for all regions of interest (ROIs) during prediction. However, not all ROIs are identified as landslides. The visualization depicts red clusters representing areas that increase the probability of a certain ROI being classified as landslides, while blue clusters signify the opposite effect. (Source: Image © 2024 Airbus)

These overpredictions are likely due to foreshortening effects in the ascending imagery, which are more pronounced in steep terrains.





The *diffVV* background in the figure provides additional context by emphasizing significant changes between pre- and post-event imagery, aiding in the identification and understanding of these false positives. Finally, we explore the interpretability of the model predictions for the Haiti event, as depicted in Figure 5. Here we present the SHAP values, which reveal how individual pixels contribute to the final detection outcomes. By averaging pixel contributions across all four image bands, areas that increase or decrease the likelihood of a region being classified as a landslide can be identified. The model exhibited

overpredictions in certain regions, particularly where model attention was directed towards riverbeds filled with sediments.

## 5   Discussion

Few rapid assessment tools and models are currently available for effective deployment in response to MLEs. Among these, only a limited number of published resources exist, with most comprising datasets intended for training generalized landslide detection models. Examples include datasets designed by Meena et al. (2023), Xu et al. (2024), and Fang et al. (2024). These

resources, as well as the tools evaluated by Amatya et al. (2023), cover both optical and SAR-based rapid assessment approaches (Handwerger et al., 2022). The SAR-based rapid assessment tool implemented by Handwerger et al. (2022) stands out for its direct applicability to new events, aligning with our research objectives. Hence, we draw a direct comparison with their methodology. While their method focuses on all surface changes and does not consider SAR geometric distortions, ours specifically targets landslide-related alterations. This distinction is crucial because post-earthquake amplitude alterations can

stem from various factors besides landslides (Xu et al., 2022). Additionally, the combination of data from both ascending and descending orbits in Amatya's method inevitably includes geometric distortions arising from the two viewing angles. This integration can compromise the accurate detection of small surface changes in mountainous areas. Moreover, our models' performance does not improve with an increase in the number of pre-event temporal stacks, contrasting with findings reported by Handwerger et al. (2022). Increasing the difference between pre- and post-event stacks beyond a certain point decreases

model performance. This tradeoff arises because larger stacks, though effective in mitigating noise, can introduce inconsistencies due to varying durations of physical processes. For example, the 12-12 case results in lower scores due to insufficient noise mitigation, while the 60-12 case achieves higher scores. However, the 365-12 case shows limited improvement, as the noise reduction saturates and physical process sampling becomes too large compared to the 12-day post-event buffer. This pattern also explains why 24-24 outperforms 60-24 and why 48-48 achieves higher scores than 60-48. Other rapid assessment

datasets and tools, such as those by Meena et al. (2023), Xu et al. (2024), Fang et al. (2024), and methods evaluated by Amatya et al. (2023), including HazMapper (Scheip and Wegmann, 2021) and ALADIM (Deprez et al., 2022), rely on optical data and employ diverse techniques such as object-based image analysis (OBIA) (Blaschke, 2010) and deep learning segmentation models. While optical-based tools excel at accurately outlining landslide boundaries, they are hindered by cloud cover and daylight dependency (Nava et al., 2022b), rendering them less effective during disaster response when rapid access to landslide

inventories is critical. Conversely, SAR-based tools provide predictions regardless of weather conditions or time of day, albeit with reduced accuracy.





## 5.1 Insights by Spatial XAI

XAI offers invaluable insights into patch classification tasks by revealing pixel contributions to the model's decisions on a per-patch basis. This analysis uncovers spurious relationships and clarifies the features the model focuses on during decision-making. XAI plays a crucial role in SAR data analysis, especially due to its less intuitive nature compared to optical imagery. In our study, XAI provides key insights into how the model distinguishes between landslides and non-landslides. Fig. 6 illustrates concrete examples of pixel contributions. Within these examples, we present four patches where the model assigns a high probability (0.87) of belonging to the landslide class in two instances. While post-event VV imagery maintains this emphasis, post-event VH imagery often shows pixels contributing to increased landslide probability without displaying the landslides themselves. This pattern persists across all VH band cases, suggesting the model uses VH for contextual insights. Despite VV and VH having minor weights in the final model output, they play crucial roles when combined with more impactful features like diffVV and diffVH.

Fig. 6(c) illustrates a scenario where the size of the landslide is not enough to produce a distinct signature in the SAR image, resulting in an indistinguishable speckle-like "salt and pepper" effect with minimal changes in dimensions, shape, and backscatter. In such cases, the model struggles to correctly classify the patch and assigns a very low probability (0.14) of containing a landslide, despite the presence of multiple landslides within the area. Moreover, while we meticulously filter the inventory to exclude landslides occurring in areas affected by shadow and layover, we can still encounter situations where landslides are not captured by SAR, resulting in bias in the model and misclassification (e.g., Fig. 6(d)). This challenge stems from our method of calculating distortion masks, which relies on the geometric interplay between the satellite's side view and the terrain, represented by the SRTM 30 m resolution DEM. Overall, the pixel contributions associated with the diffVV and diffVH bands are promising, as the model demonstrates an ability to identify the areas where most of the landslides occur and focuses attention on the landslide-related pixels. It may be worthwhile to explore applying unsupervised clustering techniques to precisely locate these landslides within the predicted landslide patches. However, it is important to note that the location of landslide-related information in SAR imagery does not always align with the location of landslides in optical imagery due to geometric distortions, which is a current inherent limitation of SAR satellite data.

## 5.2 Supporting Arguments and Gap Analysis

Our proposed method demonstrates generalization capabilities, performing well across diverse landscapes and geographical locations (Wang et al., 2020). Our approach utilizes Google Earth Engine (GEE) for data pre-processing, thus resolving the need for specialized GIS software and leverages cloud computing capabilities such as Google Colaboratory (Yang et al., 2023) to support DNNs libraries. With minimal missed detections and overpredictions, it proves reliable even when challenged with unseen landslide events. We ensure its robustness and adaptability by training and evaluating the model across areas exhibiting a wide range of landslide occurrences and environmental variations, including distinct terrains and events. The purpose of our input sampling technique is to avoid introducing *a priori* noise and biases caused by landslides located in layover and shadowing areas. By using this method, we can train the model using clear landslide patches, thereby attempting



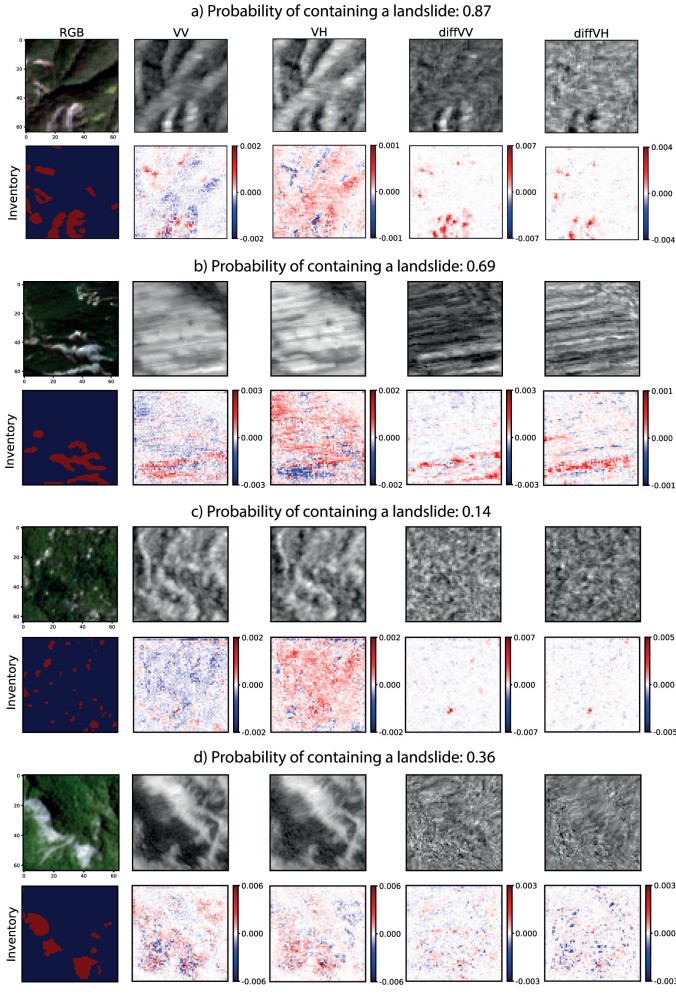

**Figure 6.** XAI pixel contribution maps. We showcase pixel contributions for each of the bands in the patches, alongside a true color image, SAR bands, landslide inventory, and SHAP pixel contributions for four patches within the mixed test set. (a) and (b) depict true positives, where (a) exemplifies a distinguishable landslide correctly detected with a high probability of belonging to the landslide class, while (b) showcases landslides accurately classified despite foreshortening effects. In contrast, (c) and (d) represent false negatives. In (c), landslides are undetected due to their small individual dimensions, illustrating a limitation of the model. (d) illustrates another case of undetected landslides, attributed to missing information in the SAR images.

to minimize the effect of sampling in mountainous regions where SAR data frequently show geometric distortions (Burrows et al., 2020). Without filtering, the model may erroneously associate specific distortions or features with landslide occurrences in situations where landslide information is not clearly represented in the SAR data. We have tested the method in a variety of environments, all of which are mainly vegetated (see Figure S1). In all the study areas, the method has proven to be effective. However, the method may perform poorly in environments like arid or semi-arid areas, where landslide inventories





are lacking. While we utilize indicators such as changes in differential maps, uncertainty arises when these changes occur for reasons not directly associated with landslides, such as water level and/or suspended material changes (Hertel et al., 2023). This uncertainty becomes particularly evident when our methodology fails to yield satisfactory results, as demonstrated by our experiences in the ascending orbit in Haiti. This challenge could be addressed if an expert reviews the images in advance, identifying any changes that may result from anomalous pre-processing or geometric acquisition errors. By filtering these

issues out prior to analysis, the impact on the results can be minimized. By removing the anomalous images from the stack, the results return to normal, highlighting the importance of careful preprocessing and quality control. Moreover, most of the inventories we used are freely available and refer to publications with peer review and validation. Nevertheless, acknowledging all the uncertainties in assessing the quality of an inventory (Guzzetti et al., 2012), even those obtained using optical images, we cannot exclude bias in the model for unseen situations. The results presented are based on the 5% pixel threshold, and

altering it could change the model's sensitivity to different landslide sizes. Additionally, the absence or minimal presence of snow in the training and calibration areas may affect our detection capabilities (see Figure S1) in regions where snow cover is present before, during, and/or right after the MLE. Another limitation lies in the model's inability to effectively detect small landslides, primarily due to Sentinel-1 data resolution constraints. Utilizing higher-resolution imagery would significantly enhance prediction accuracy. While the 5% threshold set during training helps minimize overpredictions and excludes patches

lacking landslide-related information in SAR images, it may overlook small landslides, particularly if their signals are weak or if only one small landslide is present in the area. Furthermore, our methodology cannot differentiate between different types of landslides, as such information is often unavailable in the inventories used for our research. Continuous testing is essential for mitigating uncertainties associated with unknown and unexplored terrains, poor image quality, and the presence of snow. Due to the methodology employed, conducting a sensitivity analysis on both landslide dimensions and model performance

is unfeasible. This is primarily because a single patch often encompasses multiple landslide occurrences. The models show no apparent bias given by the land cover within the areas tested (see Figure S2). However, we observed a discrepancy in the distribution of slope and aspect values when comparing the pixels within landslide scars of patches correctly predicted as landslides (*True Positives*) with those where landslides were missed (*False Negatives*). In *True Positives*, the distributions of slope and aspect align with the overall distribution observed in landslide scars across the entire training set (see Figure

S1), indicating that the model performs well when detecting landslides that align with these typical distributions. However, considering *False Negatives*, the distributions diverge, particularly in areas with high slope degrees (see Fig. 7). Slopes greater than 30°tend to introduce significant SAR backscatter distortions, which can obscure the features crucial for accurate detection. Additionally, in terms of aspect, we observe that misclassifications exhibit a peak around 160° (SE) for the descending orbit, whereas the distribution of *True Positives* remains consistent with the overall distribution, with the main peak occurring around

225° (SW). Observations indicate that the model is less likely to detect landslides occurring on slopes exceeding 30° and within the aspect range of 100° to 200° (with a peak around 160°). This aspect range is related to the Sentinel-1 descending orbit view. In this case, the combination of slopes exceeding 30° and aspects near 160° marks the point where these geometries amplify foreshortening effects in the descending orbit of the satellite (Mondini et al., 2021b).

### 5.3 Future Research Directions

Our analysis was focused on earthquake-induced MLEs. Future research will expand to include rainfall-triggered landslides. Furthermore, although our analysis focused exclusively on SAR backscatter intensity using the C-band Sentinel-1 satellite, our methodology can be reproduced or

coupled with imagery from SAR satellites operating with different radar wavelengths and resolutions, such as X and L-bands. Future SAR missions, such as NASA and ISRO's SAR initiative (NISAR), operating with S- and L-band sensors, are anticipated to offer enhanced capabilities, partic-

ularly in vegetated regions and with higher acquisition frequencies. The utilization of different wavelengths and/or polarizations could potentially provide valuable additional information, likely leading to improved performance of our models. Testing these new data would be highly intrigu-

ing, as they have the potential to enrich our understanding and further refine our methodologies for landslide detection. Exploring an alternative approach, we could begin directly from the Single Look Complex (SLC) imagery and incorporate custom filtering and co-registration techniques. This ap-

proach offers greater flexibility in creating SAR composites, enabling us to experiment with various combinations of data. For example, we could assess the performance by including differenced coherence, despite the potential decrease in spatial resolution. While coherence is known for its higher sen-

sitivity to changes compared to backscatter (Burrows et al., 2020), its inclusion may not directly enhance detection performance, but it presents a promising avenue for future investigations. Lastly, while our study focuses on a CNN, there remains significant potential for exploring additional architectures. Future research could benefit from evaluating other advanced

models, such as Visual Transformers (Liu et al., 2024) and Interactive Differential Attention Networks (Ji et al., 2024).

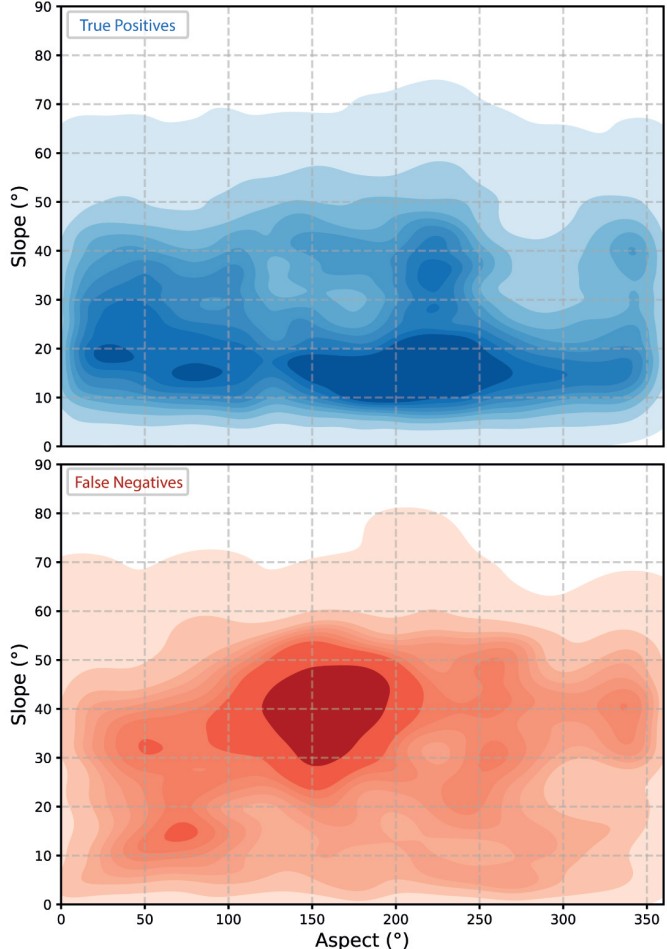

**Figure 7.** Kernel Density Estimation (KDE) of the slope and aspect values in the landslide scars, in the *True Positive* and *False Negative* predictions of the test dataset by the model trained on the descending orbit for the 60-12 temporal stack combination in the six study areas used to perform the SAR settings comparison.

## 6 Conclusions

In this study, we demonstrate the efficacy of Sentinel-1 SAR backscatter and DNNs for generalized rapid co-seismic landslide detection, introducing a data-centric approach to achieve this goal. We develop this method by using 11 earthquake-induced MLEs, comprising a total of 73 thousand landslides. We test the approach on unseen MLEs, located in Sumatra and Haiti, to validate the generalizability and applicability of the approach. We use XAI to examine the pixel contributions of the model across various SAR bands, uncovering indicators of landslide-related information, also in foreshortening. As per our current knowledge, this study represents the first evidence supporting the feasibility of applying DNNs to enable generalized landslide rapid assessment via SAR backscatter data, across diverse geographic locations. Therefore, we establish a robust foundation for future research endeavors, wherein SAR and DNNs can be harnessed to locate terrain changes in mountainous regions. The approach leverages the cloud-based capabilities of GEE and Google Colaboratory, eliminating the need for specialized software and democratizing geospatial analysis globally. Lastly, we introduce the SAR-LRA Tool, serving as an asset for swift all-weather landslide assessment, available here: https://zenodo.org/records/14898556. As reliable inventories become increasingly available, we are committed to continuously updating and refining our models and datasets to improve their accuracy and generalizability.

*Code and data availability.* The code and model weights of SAR-LRA are available at https://github.com/lorenzonava96/SAR-and-DL-for-Landslide-Rapid-Assessment and https://zenodo.org/records/14898556. As new MLE polygon inventories emerge, we will update the tool accordingly and upload the new versions in the same repository, accompanied by detailed descriptions of the modifications made.

## Appendix: Further Models Descriptions and Performance

### A.1 ResNet-Based Model

The ResNet model employs residual learning with skip connections to mitigate the vanishing gradient problem, enabling efficient training of deep networks. It uses Residual Blocks with two convolutional layers (3×3 kernels) and ReLU activation, where the input is summed with the output via skip connections. The architecture includes an initial convolutional layer, multiple Residual Blocks interspersed with MaxPooling layers, and a Dropout layer to reduce overfitting. A fully connected layer with sigmoid activation provides binary classification outputs. Optimization is performed using the Adam optimizer with a binary cross-entropy loss function, offering robust generalization for complex tasks.

### A.2 CBAM-Based Model

The CBAM model incorporates the Convolutional Block Attention Module (CBAM) to enhance feature selection via Channel and Spatial Attention (Tang et al., 2021). Channel Attention refines critical feature channels using global pooling and dense layers, while Spatial Attention emphasizes significant regions using pooled features processed through a 7×7 convolution.





The architecture consists of convolutional layers, CBAM modules, MaxPooling layers, and dense layers with Dropout for regularization. The final output uses sigmoid activation. CBAM improves feature representation with minimal computational overhead, making it effective for tasks requiring selective focus.

## A.3 Performance Comparison

**Table A1.** Performance Metrics of CNN, CBAM, and ResNet Architectures on 60-12 VV_VH Dataset.

| Model | Accuracy (%) | Precision (%) | Recall (%) | F1-score (%) |
|---|---|---|---|---|
| CNN | $96.04 \pm 0.2$ | $81.58 \pm 2.4$ | $80.17 \pm 2.5$ | $80.94 \pm 1.3$ |
| CBAM | $95.92 \pm 0.2$ | $80.75 \pm 2.8$ | $80.41 \pm 3.1$ | $80.47 \pm 0.8$ |
| ResNet | $96.43 \pm 0.2$ | $84.17 \pm 3.4$ | $81.94 \pm 2.3$ | $\mathbf{82.53 \pm 0.9}$ |

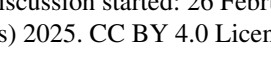



**Table A2.** Median and Standard Deviation of Accuracy, Precision, Recall, and F1-Score for VV_VH Datasets.

| Orbit | Stacks | Accuracy (%) | Precision (%) | Recall (%) | F1-score (%) |
|-------|--------|--------------|---------------|------------|--------------|
| Ascending | 365-12 | 95.48 ± 0.4 | 78.98 ± 4.0 | 77.16 ± 4.6 | 78.30 ± 2.3 |
| | 240-12 | 95.37 ± 0.4 | 81.13 ± 2.9 | 74.80 ± 4.8 | 77.30 ± 2.4 |
| | 120-12 | 95.62 ± 0.4 | 79.90 ± 3.0 | 78.23 ± 3.7 | 78.70 ± 2.3 |
| | 60-12 | 96.04 ± 0.2 | 81.58 ± 2.4 | 80.17 ± 2.5 | 80.94 ± 1.3 |
| | 12-12 | 95.31 ± 0.5 | 82.49 ± 5.1 | 72.60 ± 4.0 | 75.90 ± 1.9 |
| | 24-24 | 95.97 ± 0.4 | 83.61 ± 4.1 | 78.37 ± 3.0 | 80.10 ± 1.4 |
| | 60-24 | 96.23 ± 1.7 | 82.83 ± 6.8 | 80.05 ± 1.6 | 81.72 ± 1.4 |
| | 48-48 | 96.41 ± 2.0 | 82.55 ± 10.0 | 83.20 ± 11.6 | 83.03 ± 10.7 |
| | 60-48 | 96.29 ± 1.3 | 81.78 ± 6.7 | 83.74 ± 5.1 | 82.42 ± 5.5 |
| | 60-60 | 96.43 ± 1.5 | 81.27 ± 6.6 | 83.77 ± 1.2 | 83.24 ± 9.3 |
| Descending | 365-12 | 95.52 ± 0.4 | 79.79 ± 4.0 | 75.73 ± 3.4 | 77.54 ± 1.4 |
| | 240-12 | 95.41 ± 0.8 | 81.64 ± 6.0 | 73.46 ± 5.4 | 76.73 ± 3.4 |
| | 120-12 | 95.71 ± 0.4 | 82.01 ± 4.0 | 75.51 ± 2.0 | 78.73 ± 1.4 |
| | 60-12 | 95.52 ± 0.6 | 83.76 ± 5.3 | 72.53 ± 6.8 | 78.05 ± 3.5 |
| | 12-12 | 95.36 ± 0.3 | 83.90 ± 2.9 | 71.11 ± 3.9 | 76.30 ± 1.9 |
| | 24-24 | 95.97 ± 0.4 | 83.61 ± 4.1 | 78.37 ± 3.0 | 80.10 ± 1.4 |
| | 60-24 | 95.78 ± 0.4 | 79.53 ± 4.0 | 79.22 ± 2.8 | 79.71 ± 1.4 |
| | 48-48 | 96.41 ± 1.9 | 82.55 ± 10.2 | 83.20 ± 11.6 | 83.03 ± 10.8 |
| | 60-48 | 96.34 ± 0.4 | 81.66 ± 2.6 | 84.04 ± 2.5 | 82.80 ± 1.7 |
| | 60-60 | 96.47 ± 2.2 | 82.56 ± 11.2 | 83.87 ± 15.6 | 83.36 ± 13.6 |





**Table A3.** Median and Standard Deviation of Accuracy, Precision, Recall, and F1-Score for VV Datasets.

| Orbit | Stacks | Accuracy (%) | Precision (%) | Recall (%) | F1-score (%) |
|-------|--------|--------------|---------------|------------|--------------|
| Ascending | 365-12 | 93.57 ± 1.4 | 71.26 ± 8.1 | 65.80 ± 4.6 | 67.85 ± 4.0 |
| | 240-12 | 93.55 ± 1.2 | 72.18 ± 8.6 | 63.50 ± 4.2 | 66.51 ± 3.2 |
| | 120-12 | 93.29 ± 1.3 | 69.88 ± 8.0 | 66.47 ± 5.0 | 67.33 ± 4.5 |
| | 60-12 | 91.59 ± 2.4 | 57.95 ± 17.2 | 65.74 ± 14.1 | 60.51 ± 13.7 |
| | 12-12 | 93.39 ± 0.8 | 74.68 ± 7.9 | 56.46 ± 3.8 | 64.03 ± 3.1 |
| | 24-24 | 94.41 ± 1.5 | 76.37 ± 9.1 | 71.68 ± 4.3 | 73.03 ± 4.2 |
| | 60-24 | 94.07 ± 1.2 | 71.35 ± 6.9 | 74.41 ± 5.2 | 71.89 ± 4.0 |
| | 48-48 | 95.31 ± 0.8 | 77.22 ± 5.4 | 78.18 ± 3.5 | 77.26 ± 2.7 |
| | 60-48 | 94.76 ± 1.7 | 74.81 ± 12.6 | 77.56 ± 15.8 | 76.03 ± 15.1 |
| | 60-60 | 95.41 ± 0.3 | 77.70 ± 3.1 | 77.87 ± 3.4 | 77.64 ± 1.5 |
| Descending | 365-12 | 93.49 ± 0.8 | 73.38 ± 5.9 | 58.90 ± 9.7 | 65.50 ± 7.1 |
| | 240-12 | 93.44 ± 0.7 | 70.48 ± 5.9 | 64.34 ± 4.9 | 67.25 ± 3.1 |
| | 120-12 | 93.85 ± 0.8 | 73.49 ± 6.3 | 64.51 ± 4.6 | 68.04 ± 2.8 |
| | 60-12 | 93.81 ± 1.2 | 75.99 ± 9.2 | 61.69 ± 5.9 | 67.00 ± 4.0 |
| | 12-12 | 92.56 ± 0.6 | 68.10 ± 6.0 | 53.91 ± 5.1 | 61.66 ± 2.4 |
| | 24-24 | 93.80 ± 0.9 | 72.71 ± 6.9 | 66.29 ± 4.3 | 69.28 ± 3.1 |
| | 60-24 | 93.67 ± 1.2 | 72.52 ± 7.8 | 67.71 ± 6.1 | 68.89 ± 5.1 |
| | 48-48 | 95.37 ± 0.3 | 79.38 ± 3.6 | 75.53 ± 3.4 | 76.93 ± 1.2 |
| | 60-48 | 94.83 ± 0.3 | 74.77 ± 3.5 | 76.48 ± 4.1 | 75.23 ± 1.5 |
| | 60-60 | 95.30 ± 0.6 | 78.58 ± 5.49 | 75.98 ± 4.1 | 76.88 ± 1.7 |





*Author contributions.* L.N. contributed to the conceptualization, design of the research, formal analysis, data curation, methodology, inves-
tigation, and drafting of the original manuscript, as well as review and editing. A.C.M contributed to the conceptualization, design of the
research, investigation, and drafting of the original manuscript, as well as review and editing. K.B. and C.F. contributed to the formal analysis,
investigation, and drafting of the original manuscript, as well as review and editing. O.M., A.N., and F.C. contributed to the conceptualization
and review and editing of the manuscript.

*Competing interests.* The authors declare no competing interests.

*Acknowledgements.* The inventories utilized in our analysis primarily originate from open-access repositories, with a notable emphasis on the
"Landslide Inventories" repository provided by USGS (accessible at ). We extend our sincere appreciation to the authors for their generous
contribution of data, which greatly facilitated our research efforts. This research was supported by the "The Geosciences for Sustainable
Development" project (Budget Ministero dell'Università e della Ricerca–Dipartimenti di Eccellenza 2023–2027C93C23002690001) and
by Sichuan Science and Technology Program (No. 2024JDHJ0038). Novellino was funded through the BGS International NC programme
'Geoscience to tackle Global Environmental Challenges', NERC reference NE/X006255/1. Monserrat was funded by the Spanish Grant
PID2020-116540RB-C21 funded by MCIN/AEI /10-.13039/501100011033.





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
