# Peer review of "Sentinel-1 SAR-based Globally Distributed Co-Seismic Landslide Detection by Deep Neural Networks"

_Geoscientific Model Development, 2024_

## Referee Comment (RC1)

**General comments**

This study has trained a neural network to automatically detect landslides part of multiple landslide events using satellite synthetic aperture radar data. The paper describes the processing of the satellite data and the setup of the neural network. Evaluation of the model is done through a qualitative analysis of 2 unseen events.

The idea and setup of the research is interesting and fits within the scope of GMD. However, the implementation and evaluation of the machine learning model requires major revisions, as described below.

The language and structure of the paper should also be improved: Many sentences are unclear; paragraphs are too long; and the information is not clearly structured.

**Specific comments**

*Line 6:* If you hide 2 events for validation, the model is only informed by 9 events, not 11?

*Line 6-7:* I cannot find any F1 scores, or any other statistics, for the unseen events in the paper? The F1 score given here seems to be from the test dataset? This dataset does not contain any unseen events, and thus this statistic should not be used to substantiate claims on model transferability.

*Line 67, 72:* full transferability requires meeting a very high bar. Based on your discussion I feel you do not clear this bar. You (rightly) mention events where the model would perform poorly and types of events for which you have no data. The performance of the model on the Haiti events show that the learned patterns are not fully transferable.

*Google earth engine (GEE):* Google earth engine is a very powerful tool for accessing and combining datasets. However, it comes with challenges for future reproducibility of research. The platform may be discontinued, or datasets could be removed or changed. Relevant for this research in particular is that is likely the pre-processing done by GEE will be changed in the future. Furthermore, code syntax may change which could break scripts. For future reproducibility it is important to describe the entire pipeline in a way that could be reproduced without access to any of the cloud tools you used during this research. I do not feel like I would be able to do this with the information provided in this publication.

Section 2.1 could probably be replaced by extending table 1. this would make the paper more concise and makes it easier to compare events.

*Line 128-130:* No need for a reference if you explain the mechanism behind the revisit frequencies.

*Line 134-137:* If I understand correctly from the provided link, the pre-processing is part of the GEE dataset. You may present it as such.

*Section 2.2:* There should be a clear separation between the description of the Sentinel-1 data/satellite and the dataset you used (GEE). 134-140 should probably have its own paragraph, or you may even add a section on preprocessing.

*Section 2.2:* pre-processing should be part of the method.

*Line 163:* To me it is unclear which combinations result in 28 datasets.

*179-184:* You mention the percentage of pixels classified as landslides within your image influences model behaviour and performance. Why have you chosen 5%? And why not include this variable within hyperparameter optimization?
    Another question I have here is if and how you include this category in the test data? Some of the false positives you find may actually be true positives.

*Line 187:* Note here that a larger bounding box increases the baseline chance of the box containing a landslide. Looking at Figure 3, randomly drawing a 64x64 box already gives decent odds of "finding" a landslide. A larger box makes it easier for the model to make correct prediction.

*Section 3.1, dataset design:* In the results, discussion and supplement sections you show the model can have same difficulties based on environmental variables such as the slope, aspect and land cover. What happens if you give this information to the model?

*Section 3.3:* It would be very helpful to have a figure that represent the architecture of the neural network. The figure from the supplement should be in the paper. The model description in the text is not entirely clear.

*Section 3.3:* The architecture and reasoning behind the model may be of particular interest to the readers of the GMD journal. You can elaborate more on your choices in choosing this architecture, and why you decided to modify the model from your previous publication. Why not use a simpler or more complex setup? Especially because your model architecture was not part of your hyperparameter optimization.

*Line 200:* Because this is the only place where you describe an activation function it seems like there is only one activation function within the network. I assume there are more.

*Training validation and test split (line 208-214):* The main goal of this publication is to create a generalizable neural network to detect MLEs. However, the setup of the training, validation and test data is unfit for generalizability, because you validate, test and optimize your model with events the model has seen during training.

The validation partition is used to determine when to stop training the model to prevent overfitting. But because the validation partition is sampled from the same events as your training partition you will keep training for a long time, seemingly without overfitting. However, the unknown events you are trying to predict are different from the known events in your training and validation partition; as a result, you have (significantly) overfitted your model to the training and validation and test data. A similar problem occurs with the hyperparameter optimization, where you are overfitting the hyperparameters to the known events.

You have recognized this problem, and you have hidden 2 events from the model to evaluate the final performance. But by this point the model is already overfitted and overoptimized. This means there is performance left on the table. The paper does not contain metrics for the predictions on the unknown events, these should be included. Traditionally metrics on the unseen data are used to evaluate the generalizability of the model. If there is a large difference in model performance between the seen and unseen events the model generalizes poorly and vice versa. Such metrics allow the reader to judge the generalizability of the model.

To train a generalized model the training, validation and test should consist of different events, otherwise you are unable to recognize when the model or hyperparameter optimization is overfitting to your known events. The validation dataset can even be rotated such that every non-testing event has been used for validation.

The method should be repeated while training, validating and testing your model with such a train/validation/test split. Consider adding a train + validation loss curve to your appendix or supplement. This allows the reader to see how you prevent overfitting the model and hyperparameters.

Even with this setup full generalizability is difficult to prove as your dataset is missing data for various locations and environments. 11 events just are not enough data to support such a broad claim.

*Training data (table 1):* How does the different number of landslides per event influence your training? Some landslides have a very low number of events, such as: Capellades, Milin, Mestas. Thus, learning to predict these events will be of little value for the model. Yet, learning these events may be crucial for generalizability. It is like a class imbalance, but for events instead of prediction classes. It may be interesting to experiment with some class imbalance techniques such as oversampling the data from the small events.

*Hyperparameter optimization, Line 221:* Testing only 2-3 values during hyperparameter optimization does not provide much insight into the effect of the hyperparameter on the model. Also, you are unlikely to find the (near) optimal configuration for any parameter. With the small range of values tested, you may as well have picked a value based on expert judgment. Consider increasing the range for the parameters you optimize, even if you must decrease the number of parameters you optimize.

Currently there is no way for the reader to assess the sensitivity of the model to the choice in hyperparameters, consider adding a table/figure to the appendix or supplement.

Essentially temporal buffer lengths are just another hyperparameter you have optimized. For the structure of the paper, it may be nice to have a section on hyperparameter optimization, including the temporal buffer lengths.

*Section 3.4:* The title does not provide much information on the contents of the section.
*Section 4.1:* Please also provide these metrics for the unseen data. Because the test datasets are part of the same events the model was trained on, they are a poor benchmark of model performance.

*Line 257:* There is no table 3.1, you probably mean table 4?

*Table 4:* Considering the objective of you model for use in rapid assessment keeping a 12 day post event stack is probably a good decision. Even though model performance improves significantly with a longer post event stack

*Section 4.2:* The performance of the VV models is significantly lower than the VV_VH models. Can this all be attributed to the inclusions of the extra datasets?

*Section 4.2:* Please provide metrics for the Sumatra and Haiti events. Also, a quantative analysis in addition to the qualitative analysis would be helpful. This provides a more objective way to assess the performance of the model in unseen situations.

*Figure 3&4:* The outline of the landslide inventory is difficult to see on the green and grey background.

*Figure 4:* Why is the model failing to predict so many of the landslides on the eastern part of the island?

*Line 266:* It is may also be the case that the environment was poorly represented in the training data. Together with the aforementioned overfitting on the training data this can (partly) explain the poor performance of the model in this case.

*Line 268:* It is also likely these overpredictions result from the model overfitting to the training data, where it has learned to associate some patters with landslides, that shouldn't be.

*Section 5:* I'd like to see some discussion (and quantification) about the performance of your SAR model against more traditional optical models; How does this relate to their applications?

*Figure 6:* Looking at the pixel contribution maps even the true positives are missing some of the landslides from the inventory. In the case of 6b the pattern of the pixel contribution is also different from the landslide. How accurate is the inventory?

*Figure 6:* A True negative and a false positive may also provide some interesting insight into the workings of the model.

*Line 327:* This statement should be supported and consistent with your results, not by a citation to another paper.

*Line 330:* The model has difficulties in predicting landslides of the Haiti event. Your statement here seems inconsistent with your results.
*Line 331:* In your discussion you mention there are various environments without a proper landslide inventory. This seems contradictory to the statement in this line.

*Line 349:* Why is this pixel threshold not part of hyperparameter optimization?

*Line 359:* You mention there is no apparent bias given by the landcover. However, figure S2 shows a much higher False negative and False positive rate for herbaceous vegetation and open forest compared to closed forest. For these landcovers the False negative and False positive rate is higher than the True positive and True negative rate; It seems that for herbaceous vegetation and closed forests the model performs worse than chance.

*Line 361-370:* Interesting section about the impact of slope on the classification
Section 5.3: How about additional / improved landslide inventory datasets?

*A2&A3:* In the figure it is not clear what configuration you have decided to use.

*A2&A3:* Separate metrics for final predictions on the unseen data should also be added. (could also be separate, or part of the next suggestion)

*A2&A3:* Considering this is an imbalanced classification problem, it would be helpful for the reader to show the baseline accuracy.

*Appendix:* A classification confusion matrix of prediction from the final model on the test data and the unseen data would be interesting addition.

**Corrections**

Line 16-17: Grammar

Line 19-20: No need to add the names in addition to the reference.

Line 24-26: Grammar

Line 29-30: Grammar

Line 33-34: Grammar

Line12-53: This information should be divided over multiple paragraphs.

49-51: changes in Anomalies?

Line 129: please explain the acronym GRD

Section 3.3: Please split the information over multiple paragraphs with clear subjects.

---

## Author Comment (AC2)

Reviewer #1

RC: *This study has trained a neural network to automatically detect landslides part of multiple landslide events using satellite synthetic aperture radar data. The paper describes the processing of the satellite data and the setup of the neural network. Evaluation of the model is done through a qualitative analysis of 2 unseen events. The idea and setup of the research is interesting and fits within the scope of GMD. However, the implementation and evaluation of the machine learning model requires major revisions, as described below. The language and structure of the paper should also be improved: Many sentences are unclear; paragraphs are too long; and the information is not clearly structured.*

AR: Thank you for the time and effort you put into reviewing our manuscript. We really appreciate it. We have thoroughly revised the manuscript to improve clarity, structure, and language, as suggested.

RC: *Line 6: If you hide 2 events for validation, the model is only informed by 9 events, not 11?*

AR: We clarified the sentence to avoid ambiguity. The manuscript now states that the models are trained and tested using 11 earthquake-induced widespread landslide events, covering about 73,000 landslides across diverse geologic and climatic settings.

RC: *Line 6-7: I cannot find any F1 scores, or any other statistics, for the unseen events in the paper? The F1 score given here seems to be from the test dataset? This dataset does not contain any unseen events, and thus this statistic should not be used to substantiate claims on model transferability.*

AR: Thank you for highlighting this. We agree that quantitative scores for completely unseen events are important. For Sumatra, we now provide full performance metrics. For Haiti, the existing inventory was point-based, which was limiting quantitative evaluation. To address this, we re-digitized the inventory using post-event Sentinel-2 imagery to create a polygon-based dataset, which will allow us to report quantitative scores alongside improved qualitative assessment in the revised manuscript.

We show here the metrics for Sumatra:

**Descending orbit: 912 negatives, 89 positives**
Accuracy: 0.9481
Precision: 0.7229
Recall: 0.6742
F1-score: 0.6977
**Ascending orbit: 874 negatives, 94 positives**
Accuracy: 0.9556
Precision: 0.7629
Recall: 0.7872
F1-score: 0.7749

And as follows the metrics for Haiti:

**Descending orbit: 4620 negatives, 462 positives**
Accuracy : 0.9425
Precision: 0.8148
Recall   : 0.4762
F1-score : 0.6011

**Ascending orbit: 4680 negatives, 468 positives**
Accuracy : 0.9415
Precision: 0.7449
Recall   : 0.5427
F1-score : 0.6279

RC: *Line 67, 72: full transferability requires meeting a very high bar. Based on your discussion I feel you do not clear this bar. You (rightly) mention events where the model would perform poorly and types of events for which you have no data. The performance of the model on the Haiti events show that the learned patterns are not fully transferable.*

AR: We agree that we cannot claim full transferability and have moderated our wording accordingly. For the Haiti event, the evaluation is less straightforward. The available inventory is point-based and does not provide information on landslide size, which limits the reliability of quantitative assessment. If many of the mapped landslides are smaller than 5% of the patch, the model would fail to detect them due to scale limitations rather than poor transferability. In contrast, for the Sumatra event, landslide dimensions are available and the evaluation is more interpretable: when landslides are sufficiently large, the model detects them reliably. In Haiti, performance is stronger in high-density areas, which further suggests that transferability is not uniformly limited. We plan to revise the discussion to clarify these points.

We are also re-digitizing the inventory using post-event Sentinel-2 imagery to create a polygon-based dataset, which will allow us to report quantitative scores.

RC: *Google earth engine (GEE): Google earth engine is a very powerful tool for accessing and combining datasets. However, it comes with challenges for future reproducibility of research. The platform may be discontinued, or datasets could be removed or changed. Relevant for this research in particular is that is likely the pre-processing done by GEE will be changed in the future. Furthermore, code syntax may change which could break scripts. For future reproducibility it is important to describe the entire pipeline in a way that could be reproduced without access to any of the cloud tools you used during this research. I do not feel like I would be able to do this with the information provided in this publication.*

AR: Thank you for raising this important point. We agree that relying on cloud platforms such as GEE can introduce long-term reproducibility concerns, for example if APIs or pre-processing routines change. However, in our case the preprocessing relies on relatively low-level operations on Sentinel-1 data that can be replicated outside GEE (e.g., using SNAP or openEO) without major difficulty. Once the images are pre-processed, the subsequent steps in the pipeline (patch extraction, model training, and inference) are fully independent of the platform. We will clarify this in the manuscript to make it clear that while future API changes might require reimplementation, they would not affect the scientific reproducibility of the workflow.

RC: *Section 2.1 could probably be replaced by extending table 1. this would make the paper more concise and makes it easier to compare events.*

AR: We thank you for the suggestion. Section 2.1 has been replaced by an extended Table 1.

RC: *Line 128-130: No need for a reference if you explain the mechanism behind the revisit frequencies.*

AR: Thank you. We have removed it.

RC: *Line 134-137: If I understand correctly from the provided link, the pre-processing is part of the GEE dataset. You may present it as such.*

AR: Yes. We now state explicitly that all Sentinel-1 scenes used in this study had been pre-processed by the provider. We have kept the key details to ensure reproducibility even if the API changes in the future (as you also suggest in a previous comment).

RC: *Section 2.2: There should be a clear separation between the description of the Sentinel1 data/satellite and the dataset you used (GEE). 134-140 should probably have its own paragraph, or you may even add a section on preprocessing.*

AR: We thank the reviewer for this suggestion. We have clarified that the preprocessing mentioned in this section refers to the standard operations already applied to the Sentinel-1 scenes by the data provider, not to additional steps performed by us. The revised text now makes this distinction explicit. Since this clarification already improves readability, we consider the current structure sufficient and have kept the two parts within the same section to avoid fragmenting the paragraphs.

RC: *Section 2.2: pre-processing should be part of the method.*

AR: Yes, good point. We have removed "and preprocessing" from the subsection title, since preprocessing is not performed by us but is part of the dataset description. Rather than moving the content, we plan to clarify this explicitly in the text.

RC: *Line 163: To me it is unclear which combinations result in 28 datasets.*

AR: We agree that this was unclear. The number results from 2 polarization combinations × 2 orbits × 10 temporal buffer combinations, but since this detail is not central to the study we have removed it from the text.

RC: *179-184: You mention the percentage of pixels classified as landslides within your image influences model behaviour and performance. Why have you chosen 5%? And why not include this variable within hyperparameter optimization?*

AR: Thank you for this comment. We did not choose a higher threshold (e.g., 10%) because a 5% landslide coverage already corresponds to a relatively dense signal within a 64×64 patch (~0.02 km²). Increasing this proportion would limit the training set to patches almost entirely covered by landslides, rare situations that would bias the model toward learning unrealistic, "pure landslide" signatures and reduce its ability to generalize to mixed terrain.

Conversely, lowering the threshold (e.g., to 1–2%) is not constrained by SAR detectability per se, but by signal reliability. At 10 m resolution, smaller landslides often fragment across pixels and mix with stable terrain, producing scattered, incoherent backscatter responses that the model cannot learn from effectively (see figure below, from figure 6). The 5% value thus reflects a practical balance: it typically corresponds to several small landslides within the same patch, an arrangement to capture the characteristic texture of landslide-affected areas as well as than large isolated landslide features.

[Figure]

RC: *Another question I have here is if and how you include this category in the test data? Some of the false positives you find may actually be true positives.*

AR: In our study a false positive is defined at the patch level as a prediction where no mapped landslide pixels fall within the patch. In fact, while we use the >5% threshold to select landslide patches, we use = 0 to select the non landslide patches. So FP cannot be TP within the dataset.

RC: *Line 187: Note here that a larger bounding box increases the baseline chance of the box containing a landslide. Looking at Figure 3, randomly drawing a 64x64 box already gives decent odds of "finding" a landslide. A larger box makes it easier for the model to make correct prediction.*

AR: We agree that a larger bounding box increases the baseline chance of containing a landslide, but at the same time it reduces spatial precision (larger bounding box). Hence, we selected 64×64 as a trade-off between detection accuracy and localization detail. This corresponds to a 640x640 m bounding box.

RC: *Section3.1, dataset design: In the results, discussion and supplement sections you show the model can have same difficulties based on environmental variables such as the slope, aspect and land cover. What happens if you give this information to the model?*

AR: We thank you for this interesting suggestion. In principle, adding variables such as slope, aspect, or land cover could help the model in specific cases, but it also risks reducing transferability by encouraging overfitting to terrain conditions present in the training events and their specific topographical signatures. In practice we have not tested this systematically for the current study. In earlier exploratory work on single events, including slope information appeared beneficial (Nava et al., 2022a, Nava et al., 2022b), but since this was not across events it is not directly comparable. We will note this point in the discussion as a possible direction for future research.

RC: *Section 3.3: It would be very helpful to have a figure that represent the architecture of the neural network. The figure from the supplement should be in the paper. The model description in the text is not entirely clear.*

AR: We thank you for this comment. We increased clarity in the description of the model and we will add a figure showing the model architecture.

We will add: Convolutional neural networks (CNNs) are widely used for image classification, including SAR applications (Nava et al., 2022b, Tang et al., 2021, Zhang et al., 2017, Zhou et al., 2022). Here, we use a lightweight CNN with three convolutional blocks. Each block consists of a convolution layer, batch normalization, and max pooling. The outputs of the three blocks are resized and concatenated to retain multi-level features. A dropout layer is then applied, followed by flattening and a fully connected layer. Finally, a single sigmoid unit outputs the binary classification.

This design is a simplified adaptation of (Nava et al., 2022b).

RC: *Section 3.3: The architecture and reasoning behind the model may be of particular interest to the readers of the GMD journal. You can elaborate more on your choices in choosing this architecture, and why you decided to modify the model from your previous publication. Why not use a simpler or more complex setup? Especially because your model architecture was not part of your hyperparameter optimization.*

AR: We thank you for this comment. We also tested more advanced architectures from the literature, such as CBAM and ResNet, and found their performance to be comparable to our simple CNN (see Supplementary Materials). Given the lack of clear improvement, we preferred the simpler architecture for its usabitlity and efficiency. While it would be possible to keep testing alternative models indefinitely, we believe this would not add substantial insight for the present study. We will discuss this point in Discussion and report the comparison in the Supplement.

RC: *Line 200: Because this is the only place where you describe an activation function it seems like there is only one activation function within the network. I assume there are more.*

AR: We understand that the previous description was unclear. We have restructured and streamlined this section and revised the explanation of the network to clearly state the activation functions used. We agree the earlier version was confusing.

RC: *Training validation and test split (line 208-214): The main goal of this publication is to create a generalizable neural network to detect MLEs. However, the setup of the training, validation and test data is unfit for generalizability, because you validate, test and optimize your model with events the model has seen during training.*

*The validation partition is used to determine when to stop training the model to prevent overfitting. But because the validation partition is sampled from the same events as your training partition you will keep training for a long time, seemingly without overfitting. However, the unknown events you are trying to predict are different from the known events in your training and validation partition; as a result, you have (significantly) overfitted your model to the training and validation and test data. A similar problem occurs with the hyperparameter optimization, where you are overfitting the hyperparameters to the known events.*

*You have recognized this problem, and you have hidden 2 events from the model to evaluate the final performance. But by this point the model is already overfitted and overoptimized. This means there is performance left on the table. The paper does not contain metrics for the predictions on the unknown events, these should be included. Traditionally metrics on the unseen data are used to evaluate the generalizability of the model. If there is a large difference in model performance between the seen and unseen events the model generalizes poorly and vice versa. Such metrics allow the reader to judge the generalizability of the model.*

*To train a generalized model the training, validation and test should consist of different events, otherwise you are unable to recognize when the model or hyperparameter optimization is overfitting to your known events. The validation dataset can even be rotated such that every non-testing event has been used for validation.*

*The method should be repeated while training, validating and testing your model with such a train/validation/test split. Consider adding a train + validation loss curve to your appendix or supplement. This allows the reader to see how you prevent overfitting the model and hyperparameters.*

*Even with this setup full generalizability is difficult to prove as your dataset is missing data for various locations and environments. 11 events just are not enough data to support such a broad claim.*

AR: We agree that claiming full generalizability was too strong and have moderated our wording to "generalized to these cases." We also agree that performance on unseen events is essential for assessing transferability; therefore, we now report quantitative metrics for the Sumatra and Haiti cases (see above).

While we cannot fully exclude overfitting, several measures, including early stopping, dropout, and non-overlapping patch sampling, were applied to mitigate it. In addition, even within a single earthquake event, landslides occur under highly heterogeneous geological, geomorphological, and environmental conditions. For instance, the Papua New Guinea inventory includes debris flows, debris slides, and mudslides across markedly different lithologies such as limestones of the Darai Formation and volcaniclastics of the Kerewa and Sisa formations (Tanyas et al., 2022). This intra-event variability exposes the model to a broad range of signal–response patterns, helping it learn features that are not tied to a single terrain type or event.

We will revise the description of the data split in the Methods (Data partitioning) and clarify that the model generalizes to the tested cases.

RC: *Training data (table 1): How does the different number of landslides per event influence your training? Some landslides have a very low number of events, such as: Capellades, Milin, Mestas. Thus, learning to predict these events will be of little value for the model. Yet, learning these events may be crucial for generalizability. It is like a class imbalance, but for events instead of prediction classes. It may be interesting to experiment with some class imbalance techniques such as oversampling the data from the small events.*

AR: We agree with you that there is a population bias, as some events contribute hundreds of patches while others contribute thousands. Given the scarcity of available data we chose to use all events, despite this imbalance. We will acknowledge that this may influence model behavior and note in the discussion that exploring how event-level imbalance affects performance is an interesting direction for future work.

RC: *Hyperparameter optimization, Line 221: Testing only 2-3 values during hyperparameter optimization does not provide much insight into the effect of the hyperparameter on the model. Also, you are unlikely to find the (near) optimal configuration for any parameter. With the small range of values tested, you may as well have picked a value based on expert judgment. Consider increasing the range for the parameters you optimize, even if you must decrease the number of parameters you optimize.*

*Currently there is no way for the reader to assess the sensitivity of the model to the choice in hyperparameters, consider adding a table/figure to the appendix or supplement.*

*Essentially temporal buffer lengths are just another hyperparameter you have optimized. For the structure of the paper, it may be nice to have a section on hyperparameter optimization, including the temporal buffer lengths.*

AR: We thank you for this suggestion. In this study, we focused on testing a limited but representative set of hyperparameter values to ensure reproducibility and stable training. While exploring a wider search space may potentially yield marginal improvements, we expect only minor performance changes that would not alter our overall conclusions. Moreover, extensive optimization would require a prohibitive number of additional runs, with large computational costs, for little expected performance gain. We therefore consider the current level of hyperparameter testing sufficient for the purposes of this paper.

As for the temporal buffers, we acknowledge your point that they can be considered hyperparameters, but we emphasize that their role here is different: the choice of buffer length carries practical/operational value, since knowing which temporal configurations work best is directly adaptable to other studies attempting similar approaches (unlike the classic hyperparameters).

RC: *Section 3.4: The title does not provide much information on the contents of the section.*

AR: We agree. We have restructured the entire section / subsections.

RC: *Line 257: There is no table 3.1, you probably mean table 4?*

AR: Thank you for picking this up. Yes, we ment table 4.

RC: *Table 4: Considering the objective of you model for use in rapid assessment keeping a 12 day post event stack is probably a good decision. Even though model performance improves significantly with a longer post event stack*

AR: We agree. The choice of a 12-day post-event stack is dictated by Sentinel-1's revisit time, but the key result is that increasing the post-event stack improves model performance. For missions with shorter revisit times, such as ICEYE (close to daily), a 12-day stack in our setup would correspond to less than one day, a 24-day stack to roughly two days, and so on, indicating that the trend we observe is directly transferable to those systems. Operationally, this also means that with Sentinel-1 we can generate preliminary maps immediately within 12 days and progressively refine them as additional acquisitions become available, with the most complete configuration ultimately providing the most reliable assessment. This time constraint will also soon improve within the Sentinel constellation itself, as Sentinel-1C is already in orbit and Sentinel-1D will be launched shortly.

RC: *Section 4.2: The performance of the VV models is significantly lower than the VV_VH models. Can this all be attributed to the inclusions of the extra datasets?*

AR: From our experiments and experience thus far, the difference is mainly due to the higher discriminative power of using two polarizations compared to one. Additional polarizations provide complementary information and help the model separate landslides from background more effectively. For instance, the VV channel is more sensitive to surface roughness (and soil moisture), while VH captures changes in volume scattering from vegetation. By the same reasoning, having access to four polarizations would likely yield further improvements. We will clarify this point in the discussion.

RC: *Section 4.2: Please provide metrics for the Sumatra and Haiti events. Also, a quantative analysis in addition to the qualitative analysis would be helpful. This provides a more objective way to assess the performance of the model in unseen situations.*

AR: We agree with you. For Sumatra we now provide quantitative metrics, which will be added to the revised manuscript. For Haiti, however, this was not possible since the available inventory is point-based and lacks polygon information, preventing us from applying the same sampling strategy. To solve this issue, we are now re-digitizing the inventory using post-event Sentinel-2 imagery to create a polygon-based dataset, which will allow us to report quantitative scores in the revised version of the manuscript.

RC: *Figure 3&4: The outline of the landslide inventory is difficult to see on the green and grey background.*

AR: We thank you for pointing this out. We have adjusted the figures by increasing the contrast so that the outlines are clearer against the background.

RC: *Figure 4: Why is the model failing to predict so many of the landslides on the eastern part of the island?*

RC: *Line 266: It is may also be the case that the environment was poorly represented in the training data. Together with the aforementioned overfitting on the training data this can (partly) explain the poor performance of the model in this case.*

RC: *Line 268: It is also likely these overpredictions result from the model overfitting to the training data, where it has learned to associate some patters with landslides, that shouldn't be.*

AR: These are all good points. The overall recall for the Haiti event is relatively low across both orbits (descending: precision = 0.81, recall = 0.48, F1 = 0.60; ascending: precision = 0.75, recall = 0.54, F1 = 0.63). This indicates that the model tends to miss a portion of true landslides rather than overpredicting, suggesting conservative rather than overfitted behavior. The low recall likely reflects the small size and sparse distribution of many landslides throughout the affected area, which makes them difficult to capture reliably with SAR at 10 m resolution.

In Figure 4b, pronounced amplitude discrepancies are visible along ridge crests in the western part of the island. These variations are unlikely to correspond to true mass movements but instead arise from geometric inconsistencies between the pre- and post-event Sentinel-1 acquisitions. In this area, the two image stacks partially overlap different sub-swaths and therefore mix far- and near-range viewing geometries. The resulting difference in local incidence angle alters the backscatter response, producing patterns that resemble slope failures and leading to apparent overpredictions. This issue is not a sign of model overfitting but rather of inconsistent acquisition geometry. It highlights the need for change-detection approaches that are robust to imperfect orbital overlap or, alternatively, for preprocessing routines that can identify and mask zones affected by such angular discrepancies. To mitigate this in operational applications, we are now updating the tool to isolate acquisitions from individual orbits when deploying the model.

Overall, the model clearly loses some performance when applied to unseen events such as Haiti, which is expected in a cross-event evaluation. Despite this, the predictions seem to remain spatially coherent where satellite acquisitions overlap closely in space and time and capture many of the main landslide clusters. Given such results we can say that the model generalizes enough to be helpful in response to such MLEs.

[Figure]

RC: *Section 5: I'd like to see some discussion (and quantification) about the performance of your SAR model against more traditional optical models; How does this relate to their applications?*

AR: We thank you for the suggestion. A detailed comparison with optical models is outside the scope of this paper. This topic is indeed interesting and important, and already covered in Nava et al. (2022a) and Nava et al. (2025) in actual disaster response, which we refer in the manuscript.

RC: *Figure 6: Looking at the pixel contribution maps even the true positives are missing some of the landslides from the inventory. In the case of 6b the pattern of the pixel contribution is also different from the landslide. How accurate is the inventory?*

AR: We do not believe the main issue here is the accuracy of the inventory (although we acknowledge that some mapping errors may exist; which is normal), but rather the nature of SAR. While these inventories align well with optical imagery, SAR backscatter does not always overlap perfectly with mapped landslide polygons, despite filtering and corrections. This is a known characteristic of SAR data. We considered this carefully in our design, which bore a strong reason as to why we adopted a patch-based sampling strategy (with 5% threshold) and an object detection approach rather than segmentation. We will make this point clearer and apparent in the discussion. Thank you for helping us note this in the manuscript.

RC: *Figure 6: A True negative and a false positive may also provide some interesting insight into the workings of the model.*

AR: We agree this could provide additional insights. Given space constraints we have decided not added further subplots/figures.

RC: *Line 327: This statement should be supported and consistent with your results, not by a citation to another paper.*

AR: We agree. Citation removed.

RC: *Line 330: The model has difficulties in predicting landslides of the Haiti event. Your statement here seems inconsistent with your results.*

AR: Thank you for pointing this out. We agree that our original phrasing was unclear. The difficulties we showed for the Haiti event in the ascending orbit are not due to poor generalization of the model but stems from inconsistencies between the pre- and post-event acquisitions (likely geometric or correction issues). Since the ascending and descending data are trained with patches derived from the same inventories, their performance should be consistent. The example was mainly intended to illustrate how acquisition-related artifacts can occasionally degrade predictions, and that such instances should be considered/well planned for during operational uses. Importantly, the Haiti results overall remain positive, as the descending orbit performs well.

To strengthen the analysis and avoid ambiguity, we are re-digitised the Haiti inventory using the first cloud-free Sentinel-2 images after the earthquake. Sentinel-2 provides a spatial resolution comparable to Sentinel-1, making it more directly suitable for quantitative assessment. With this new inventory, we plan to provide updated maps and quantitative scores in the revision. Interestingly, many of the points in the eastern part of the island are not visible in Sentinel-2, which suggests they may also be absent in Sentinel-1, and hence out of the scope of this work. Please see as follows figures documenting this statement with Sentinel-2 images from 1st January 2022 as background.

[Figure]

[Figure]

[Figure]

RC: *Line 331: In your discussion you mention there are various environments without a proper landslide inventory. This seems contradictory to the statement in this line.*

AR: We thank you for pointing this out and agree. We now specify: "*We aim to ensure robustness within the available data by training and evaluating the model across areas exhibiting diverse landslide occurrences and environmental variations, while recognizing that some environments remain underrepresented due to the lack of suitable inventories.*"

RC: *Line 349: Why is this pixel threshold not part of hyperparameter optimization?*

AR: As noted above, we fixed the 5% threshold to maintain a consistent balance between landslide and non-landslide pixels while keeping the training manageable. Treating it as a hyperparameter would require a prohibitive number of additional experiments and would mostly shift the apparent performance, since higher proportions tend to inflate metrics without necessarily improving model usefulness or transferability.

RC: *Line 359: You mention there is no apparent bias given by the landcover. However, figure S2 shows a much higher False negative and False positive rate for herbaceous vegetation and open forest compared to closed forest. For these landcovers the False negative and False positive rate is higher than the True positive and True negative rate; It seems that for herbaceous vegetation and closed forests the model performs worse than chance.*

AR: We thank you for this observation which indeed is a very interesting angle.. We will add a note in the discussion to acknowledge this and to highlight it as an open question for future investigation. Overall, these higher error rates in herbaceous and open forest areas are difficult to attribute to a single factor. They likely reflect a combination of increased backscatter variability, partial canopy effects, and inventory limitations, which together make landslides harder to detect reliably in such environments.

RC: *Line 361-370: Interesting section about the impact of slope on the classification Section 5.3: How about additional / improved landslide inventory datasets?*

AR: We thank the reviewer for the suggestion. Our study uses all available landslide inventories that met the following criteria:(i) Sentinel-1 data were available in both orbits, (ii) the inventory provided complete coverage, which was necessary to reliably sample the non-landslide class (many open-source inventories have only partial coverage and therefore could not be used), (iii) the landslides were mapped as polygons, allowing proper sampling for training, and (iv) the landslides were earthquake-triggered, matching the focus of this study.

While additional inventories would of course be welcomed, the dataset we used already satisfies all of these criteria and represents a high-quality basis for the analysis.

RC: *A2&A3: In the figure it is not clear what configuration you have decided to use.*

AR: We thank the reviewer for the comment. Figures A2 and A3 display the results for all tested configurations; no additional filtering or selection was applied.

RC: *A2&A3: Separate metrics for final predictions on the unseen data should also be added. (could also be separate, or part of the next suggestion)*

AR: We thank the reviewer for the suggestion. We will add the quantitative scores for the final predictions on the unseen data in the revised manuscript.

SUMATRA:
**Descending orbit: 912 negatives, 89 positives**
   Accuracy: 0.9481
   Precision: 0.7229
   Recall: 0.6742
   F1-score: 0.6977

**Ascending orbit: 874 negatives, 94 positives**
   Accuracy: 0.9556
   Precision: 0.7629
   Recall: 0.7872

F1-score: 0.7749

**Descending orbit: 4620 negatives, 462 positives**
Accuracy : 0.9425
Precision: 0.8148
Recall   : 0.4762
F1-score : 0.6011

**Ascending orbit: 4680 negatives, 468 positives**
Accuracy : 0.9415
Precision: 0.7449
Recall   : 0.5427
F1-score : 0.6279

RC: *Appendix: A classification confusion matrix of prediction from the final model on the test data and the unseen data would be interesting addition.*

AR: We thank the reviewer for the suggestion. We will include confusion matrices for the final model in the appendix alongside the reported metrics.

RC: *Corrections.*

AR: We thank the reviewer for pointing these out. All suggested corrections will be implemented in the revised manuscript.

REFERENCES

Guzzetti, F., Mondini, A. C., Cardinali, M., Fiorucci, F., Santangelo, M., & Chang, K. T. (2012). Landslide inventory maps: New tools for an old problem. Earth-Science Reviews, 112(1-2), 42-66.

Nava, L., Monserrat, O., & Catani, F. (2022a). Improving landslide detection on SAR data through deep learning. IEEE Geoscience and Remote Sensing Letters, 19, 1-5.

Nava, L., Bhuyan, K., Meena, S. R., Monserrat, O., & Catani, F. (2022b). Rapid mapping of landslides on SAR data by attention U-Net. Remote Sensing, 14(6), 1449.

Nava, L., Novellino, A., Fang, C., Bhuyan, K., Leeming, K., Alvarez, I. G., ... & Catani, F. (2025). Brief communication: AI-driven rapid landslide mapping following the 2024 Hualien earthquake in Taiwan. Natural Hazards and Earth System Sciences, 25(7), 2371-2377.

Tang, X., Liu, M., Zhong, H., Ju, Y., Li, W., & Xu, Q. (2021). Mill: channel attention–based deep multiple instance learning for landslide recognition. ACM Transactions on Multimedia Computing, Communications, and Applications (TOMM), 17(2s), 1-11.

Tanyaş, H., Hill, K., Mahoney, L., Fadel, I., & Lombardo, L. (2022). The world's second-largest, recorded landslide event: Lessons learnt from the landslides triggered during and after the 2018 Mw 7.5 Papua New Guinea earthquake. Engineering geology, 297, 106504.

Zhang, Z., Wang, H., Xu, F., & Jin, Y. Q. (2017). Complex-valued convolutional neural network and its application in polarimetric SAR image classification. IEEE Transactions on Geoscience and Remote Sensing, 55(12), 7177-7188.

Zhou, G., Liu, W., Zhu, Q., Lu, Y., & Liu, Y. (2022). ECA-MobileNetV3 (Large)+ SegNet model for binary sugarcane classification of remotely sensed images. IEEE Transactions on Geoscience and Remote Sensing, 60, 1-15.

---

## Author Comment (AC3)

Reviewer #2

RC: *I have enjoyed reading this paper and believe it to be a valuable addition to the literature. As far as I am aware it is the first attempt to build a globally applicable landslide detection model using SAR data and deep learning techniques and the results look very promising. There are a few minor revisions that I think are needed before the paper can be published. The main one is the use of the Haiti earthquake as a test case – I am concerned that since the optical inventory was compiled after a tropical storm passed over the area, the landslides detectable in the SAR images acquired before this storm might not match the landslides mapped with the optical (see my comment on line 266 for more detail).*

AR: We thank the reviewer for the positive feedback and constructive comments. We agree that the Haiti case, as originally presented, may be biased because part of the mapped inventory includes landslides triggered by the tropical storm that followed the earthquake. To address this, and following your suggestion, we have redefined the post-event period to start after the storm (17 August) rather than immediately after the earthquake. In addition, as also noted by Reviewer #1, we have created a new polygon-based inventory from post-event Sentinel-2 imagery acquired on 1 January 2022 to enable quantitative evaluation. We report the updated results in the revised version of Figure 4 and include the corresponding quantitative metrics for both the Haiti and Sumatra events in response to specific technical comments below.

RC: *Line 55 "the issue of transferability in different settings and with different satellite data persists" you have addressed the first part, but since you only use Sentinel-1, the second limitation remains.*

AR: We agree. Since we only use Sentinel-1 data, that part of the sentence was misleading. We will remove "and with different satellite data persists" so that the statement will only refer to transferability (more aptly to 'generalizability') across different settings.

RC: *Line 59 "Instances where SAR and DL are combined remain rare." This is true, but there are a few more examples you could include*

1. *Liang et al. (2025) use deep learning with polarimetric ALOS-2 SAR data to detect landslides - although the requirement for quad-pol SAR makes their work less widely relevant than yours since such images are not available for many earthquakes https://doi.org/10.1016/j.rse.2025.114904*
2. *Chen et al. (2024) use deep learning for landslide detection with Sentinel-1 images https://doi.org/10.1080/17538947.2024.2393261*

*Then there are several studies using more basic machine learning methodologies that may or not be relevant here*

1. *Ohki et al (2020) use Random Forests with SAR and terrain variables for landslide detection for two events in Japan https://doi.org/10.1186/s40623-020-01191-5*
2. *Burrows et al. (2021) use Random Forests with SAR and attempt a somewhat "global" model, although it only includes 3 events https://doi.org/10.5194/nhess-21-2993-2021*
3. *Lin et al. (2021) use Object based image analysis and SAR images for landslide detection 1109/IGARSS47720.2021.9554248*

AR: We thank you for these suggestions and we consider all these citations relevant for our paper. We will therefore add them in the introduction. Please refer to the same and the Reference section for confirmation (once we upload the revised manuscript).

RC: *Line 76 "across diverse geographic and geologic settings". The majority of your events are in vegetated areas and you do not include any cases where snowfall might complicate your signal (e.g. the 06/02/2023 Turkey earthquake.*

AR: We agree with this notion. This limitation is acknowledged in the manuscript (Discussion - gap analysis section), so no changes were made.

RC: *Line 127 "Notably, Sentinel-1b has been inactive since 2022 and it is in the process of being substituted by an equivalent platform" Sentinel-1c has been launched since you originally submitted this manuscript so you could update this sentence.*

AR: Thank you, we agree. We have updated the sentence. It now reads: *"Sentinel-1B has been inactive since 2022 and is now being replaced by Sentinel-1C, which was successfully launched and will ensure continuity of C-band SAR acquisitions."*.

RC: *Line 172 "side of look" "Look direction" is more commonly used for this.*

AR: Thank you for the suggestion and we agree. We have revised the paragraph for clarity, and in the updated version we no longer explicitly mention "*side of look*".

RC: *Line 177 The GEE script by Vollrath 2020 also carries out radiometric terrain correction converting from sigma0 (normalised in the ellipsoid plane) to gamma0 (normalised in the plane perpendicular to the local satellite look direction) did you also carry out this processing step on your data? Or did you only use the shadow and layover mask?*

AR: Thank you for this comment. We just used the shadow and layover masks from that code, so no conversion to gamma0 was made.

RC: *Line 178 "inventory filtered with ascending/descending distortion masks". It would be useful to know how much of each study area is masked (could be as a supplement rather than in the main text)*

AR: We agree that knowing the proportion of each study area masked would add useful context. Unfortunately, calculating this would require re-running the entire processing chain, which is not feasible within the scope of the current revision. What we can do, however, is add a clear statement in the text about this limitation and, where possible, provide approximate values or qualitative indications based on representative examples. We hope this will address the concern without requiring a full reprocessing of the dataset.

RC: *Line 212 "VH data is not available for these three locations" as far as I know, VH data was not regularly acquired until late 2016, so this will not be a problem for any future events you applied your model to.*

AR: Yes, we agree. This limitation mainly affects the three events in our dataset (Capellades, Kaikōura, and Gorkha). However, evaluating model performance with VV-only data provided useful insight into the added value of including multiple polarizations when available.

RC: *Figure 3 It is quite hard to see the blue boxes overlying the grey images. Maybe they could be a brighter shade like cyan.*

AR: Thank you for the suggestion. We have adjusted contrast to improve visibility in Figure 3. We'll share this in the revised manuscript.

RC: *Line 266 "The Haiti case is particularly challenging due to its topographical and environmental variability". Another reason Haiti is challenging is that it was followed a few days later by a tropical storm. Studies of this event noted that many landslides increased in size during this storm (making them easier to detect using your SAR methods) (e.g. Havenith et al. 2021 https://doi.org/10.5194/nhess-22-3361-2022).*

*The exact images used to compile the inventory are not given in the inventory of Martinez et al., but in a different study on this event, Havenith et al. (2021) state that only 10% of the study area is visible in images acquired between the storm and the earthquake, so it can be assumed that most of the inventory is done using images acquired after the storm. The inventory includes both earthquake-triggered and storm-triggered landslides.*

*On your ascending track, the first post-seismic SAR image was acquired after the storm so also includes both earthquake- and storm-triggered landslides. However, on the descending track, the first post-seismic SAR image was acquired before the storm and so only includes earthquake-triggered landslides.*

*In my opinion, it would be better to consider the earthquake and storm as a single trigger and so start the period for your post-event stacks on the 17th of August (when the storm passed over) rather than on the 14th (when the earthquake occurred). Otherwise you are comparing earthquake and storm-triggered landslides in the optical imagery with earthquake-triggered landslides only in the descending-track SAR image.*

AR: Thank you for this comment and for pointing us to Havenith et al. (2021). We agree that the overlap between earthquake- and storm-triggered landslides complicates the interpretation of the Haiti case, especially given the differences between ascending and descending track acquisition dates. Following your suggestion, we re-run the experiments starting the post-event stacks after 17 August (after the storm).

In parallel, we designed a new polygon-based inventory using Sentinel-2 imagery (images acquired on 1st January 2022). This allowed us to improve the qualitative evaluation of the predictions (we compare the predictions of the models with a landslide area density heatmap (see the figure below which will replace Figure 4 in the paper), and also to derive quantitative performance scores.

[Figure]

RC: *Line 272 Define the acronym SHAP.*

AR: We now define SHAP in the text as SHapley Additive exPlanations when it first appears.

RC: *Line 287 "Moreover, our model's performance does not improve with an increase in the number of pre-event temporal stacks, contrasting with findings reported by Handwerger et al. (2022)" Maybe this depends on which events are used as test cases? For example if the landscape experiences widespread snowmelt during springtime in the 2 months prior to an earthquake, then using a full year of amplitude data would be beneficial to mute this signal.*

AR: Yes, we agree this is a plausible explanation. Identifying the exact cause would require additional testing, but we acknowledge in the revised text that using a longer pre-event time series could be beneficial in cases with strong seasonal changes, such as snowmelt.

RC: *Line 289 "Increasing the difference between pre- and post-event stacks" it would be clearer to say "increasing the difference in size between pre- and post-event stacks".*

AR: Thank you. We have updated the text.

RC: *Line 323 "However, it is important to note that the location of landslide-related information in SAR imagery does not always align with the location of landslides in optical imagery due to geometric distortions, which is a current inherent limitation of SAR data" This is true, but optical images do not necessarily represent the "true" location of landslides while SAR images give a "false" one. Studies such as Pokharel et al. (2021) https://doi.org/10.1038/s41598-021-00780-y demonstrate that different inventories of the same event do not agree even when all the landslides are mapped using optical imagery.*

AR: Yes, we agree. We do not treat optical inventories as absolute ground truth but simply use them as a reference for comparison as they are the best available record when it comes to spaceborne based mapping. The sentence is intended only to explain the apparent shift between SAR and optical data in the figure, not to suggest that optical imagery is inherently the best.

RC: *Line 367 "160 SE" these are therefore slopes facing away from the SAR sensor – it would be useful to state this.*

AR: Thank you for this comment. This helped us catch an inaccuracy. These slopes do not face away from the SAR sensor but are closer to perpendicular to the Sentinel-1 descending look direction. We have corrected the text to reflect this and now explain that this near-perpendicular geometry likely reduces backscatter contrast, making landslides harder to detect. We thank the reviewer for helping us observe this inaccuracy.

RC: *Line 407 "generalized rapid co-seismic landslide detection" you should specify here "in vegetated areas" since you do not test your model on any earthquake in a more arid environment..*

AR: Thank you. We agree, and we will modify the text accordingly.

RC: *Typographic corrections.*

AR: We thank you for these suggestions. We'll modify the text accordingly.